



# Post-depositional vivianite formation alters sediment phosphorus records

Nikki Dijkstra[1,2], Mathilde Hagens[1], Matthias Egger[1,3], Caroline P. Slomp[1]

5   [1]Department of Earth Sciences - Geochemistry, Faculty of Geosciences, Utrecht University, Utrecht, Princetonplein 9, 3584 CC, the Netherlands

[2]Now at: Hoogheemraadschap De Stichtse Rijnlanden, Poldermolen 2, 3994 Houten, the Netherlands

[3]Now at: Center for Geomicrobiology, Department of Bioscience, Aarhus University, Ny Munkegade 114, 8000 Aarhus, Denmark

*Correspondence to*: Nikki Dijkstra (n.dijkstra@uu.nl)

**Abstract.** Phosphorus (P) concentrations in sediments are frequently used to reconstruct past environmental conditions in freshwater and marine systems, with high values thought to be indicative of a high biological productivity. Recent studies suggest that the post-depositional formation of vivianite, an iron(II)-phosphate mineral, might significantly alter trends in P with sediment depth. To assess its importance, we investigate a sediment record from the Bornholm Basin that was retrieved

during the IODP Baltic Sea Paleoenvironment Expedition 347 in 2013, consisting of lake sediments overlain by brackish-marine deposits. Combining bulk sediment geochemistry with micro-analysis using scanning electron microscope energy dispersive spectroscopy (SEM-EDS) and synchrotron-based X-ray absorption spectroscopy (XAS), we demonstrate that manganese- and magnesium-rich vivianite is present in the lake deposits just below the transition to the brackish-marine sediments (at 11.5 to 12 m sediment depth). In this depth interval, phosphate that diffuses down from the organic-rich, brackish-

marine sediments meets porewaters rich in dissolved iron in the lake sediments, resulting in vivianite precipitation. Results from a reactive transport model suggest that the peak in vivianite originally occurred at the lake-marine transition (9 to 10 m) and moved downwards due to changes in the depth of a sulfidization front. However, its current position relative to the lake-marine transition is stable as the vivianite and active sulfidization fronts have been spatially separated over time. Experiments in which vivianite was subjected to sulfidic conditions demonstrate that incorporation of manganese or magnesium in vivianite

does not affect its susceptibility to sulfide-induced dissolution. Our work highlights that post-depositional vivianite formation has the potential to strongly alter sedimentary P records particularly in systems that are subject to environmental perturbation, such as a change in primary productivity, which can be associated with a lake-marine transition.

**1 Introduction**

Phosphorus (P) is a key nutrient for marine organisms as it is essential for intracellular energy, cell growth and metabolism (Tyrrell, 1999). In marine waters, dissolved phosphate (henceforth denoted as $PO_4$) is converted into particulate organic P by phytoplankton. Part of this particulate P sinks through the water column as dead organic matter and is deposited onto the

seafloor (Ruttenberg, 2003). In surface sediments below oxic bottom waters, $PO_4$ that is released by the decomposition of organic matter typically diffuses back into the water column or is bound to iron-oxyhydroxides (Fe-oxide bound P). Most Fe-oxides are dissolved by dissimilatory Fe(III) reduction or by reactions with hydrogen sulfide ($HS^-$) upon burial (Canfield et al., 1992), thereby releasing $PO_4$ to the porewater. The subsequent accumulation of porewater $PO_4$ at depth in the sediments can promote precipitation of P-bearing minerals such as apatites (authigenic calcium(Ca)-P; mainly carbonate fluor-apatite).

This so-called "sink-switching" acts as an efficient P trap in sediments and limits the diffusive flux of $PO_4$ towards the water column (Ruttenberg and Berner, 1993; Slomp et al., 1996). As a consequence, sedimentary P concentrations can be a measure of the organic matter supply to the sediment (Calvert and Pedersen, 2007; Tribovillard et al., 2006). Trends in sedimentary P with sediment depth have for instance been used as evidence that upwelling zones were more productive during interglacial stages compared to glacial stages (Ganeshram et al., 2002).

In sediments overlain by hypoxic or anoxic bottom waters (oxygen concentrations below 2 mg/L or absent, respectively), P is generally less efficiently retained in the surface sediments. This is due to the absence of Fe-oxides that can adsorb P (Slomp et al., 1996) and other redox-dependent mechanisms, such as the decreased sequestration of polyphosphates in sediments overlain by anoxic waters (Sannigrahi and Ingall, 2005) and the preferential regeneration of P from organic matter (Ingall et al., 1993; Ingall and Jahnke, 1997; Jilbert et al., 2011). Less retention of P in surface sediments can hamper build-up

of porewater $PO_4$ in the sediments and sink-switching of P to P-bearing minerals, resulting in less burial of P relative to organic carbon ($C_{org}$). Trends in the ratio of $C_{org}$ to total P ($C_{org}/P$) with sediment depth thus can reflect bottom water redox conditions. This has been demonstrated for various environments and on various timescales, ranging from the Phanerozoic Ocean (Algeo and Ingall, 2007) and the Quaternary Mediterranean (Kraal et al., 2010) to the modern Black Sea (Dijkstra et al., 2014) and Baltic Sea (Jilbert and Slomp, 2013).

Besides sedimentary P concentrations and $C_{org}$/P-ratios, the occurrence of vivianite ($(Fe_3PO_4)_2.8H_2O$) in sediments is also used in reconstructions of paleo-environmental conditions in lake and marine basins (e.g. Derkachev et al., 2007; Kalniņa et al., 2000; Taldenkova et al., 2010). These Fe(II)-phosphates can precipitate in surface sediments with sulfide-depleted porewaters that are rich in both $Fe^{2+}$ and $PO_4$ (Berner, 1981; Nriagu and Dell, 1974; Nriagu, 1972). The presence of vivianite

in sediments may therefore reflect anoxic non-sulfidic bottom waters upon sediment deposition. The large vivianite concretions (> 500 µm) in Late Glacial and Holocene sediments from the Laptev Sea are, for example, assumed to be the result of low bottom water oxygen upon sediment deposition (Taldenkova et al., 2010). Vivianite minerals that were discovered in the presence of organic matter and iron sulfides in marine Latvian sediments from the middle Pleistocene were also linked to oxygen-poor bottom waters at the time of deposition (Kalniņa et al., 2000).

Post-depositional processes during burial can, however, alter the geochemical signature of sediments. An active downward oxidation front, for instance, can reoxygenate former anoxic organic-rich sediments, resulting in a decrease in $C_{org}$/P-ratios as $C_{org}$ is remineralized while the released P is retained in the sediments (Slomp et al., 2002). Post-depositional diagenesis can also result in the precipitation of vivianite in sediments. This is known to occur below the sulfate/methane transition zone (SMTZ) in sediments, where porewater $Fe^{2+}$ can accumulate in the absence of sulfide and react with porewater

$PO_4$ (Egger et al., 2015b, 2016; Hsu et al., 2014; März et al., 2008). Vivianite can also form post-depositionally at lake-marine transitions, where upward diffusing porewater $Fe^{2+}$ from lake deposits meets $PO_4$ from the overlying organic-rich marine deposits, as recently confirmed by micro-spectroscopy in sediments from the Landsort Deep (Baltic Sea; Dijkstra et al., 2016). Post-depositional vivianite authigenesis is also assumed to have caused a distinct P maximum (~ 60 µmol/g) at the lake-marine transition in coastal Bothnian Sea sediments (Dijkstra et al., 2017). The occurrence of vivianite in sediments can thus not

always be ascribed to the environmental conditions in the surface sediments and overlying waters upon deposition.

Diagenesis can also result in the dissolution of depositional or post-depositional vivianite in sediments, as this P-bearing mineral is assumed to dissolve in the presence of even small concentrations of sulfide (Vuillemin et al., 2013). Recently, sulfide-induced dissolution of vivianite was included in a reactive transport model for deep basin sediments in the Black Sea and significantly improved the model fits for porewater $PO_4$ and $Fe^{2+}$, as well as for sedimentary P (Egger et al.,

2016). This Black Sea case study showed that diffusion of sulfide into lake sediments can result in a continuous downward

migrating vivianite front just below the sulfidization front. Over the past 7500 years, the peak in vivianite in these Black Sea sediments may have migrated downwards from its initial position at the lake-marine transition over a distance of more than two meters (Egger et al., 2016). In contrast to the Black Sea, the Landsort Deep in the Baltic Sea was only intermittently sulfidic throughout the Holocene and vivianite is preserved close to the lake-marine transition (Dijkstra et al., 2016). The

sediment depth at which post-depositional vivianite can overprint paleo-records of total P and $C_{org}$/P may thus depend on the porewater sulfide concentrations during sediment burial.

Also other factors might affect the stability and, subsequently, the presence of vivianite in sediments. In many lake and marine sediments, vivianite aggregates are enriched in magnesium (Mg) and manganese (Mn) (Dijkstra et al., 2016; Egger et al., 2015b; Hsu et al., 2014). These enrichments may reflect the substitution of $Fe^{2+}$ by $Mn^{2+}$ and $Mg^{2+}$ in the crystal structure

of vivianite (Dijkstra et al., 2016; Frost et al., 2002; Nakano, 1992). In Fe-oxides, the incorporation of cations such as $Mn^{3+}$ into the crystal structure affects the mineral stability and can result in higher acid dissolution rates of Fe-oxides (Alvarez et al., 2007). At present, it is unknown whether the substitution of $Fe^{2+}$ in vivianite crystals by other cations affects the mineral stability and its susceptibility to sulfide dissolution.

In this study, we investigate the post-depositional formation of vivianite in sediments of a hypoxic basin in the Baltic

Sea, the Bornholm Basin. Combining various types of geochemical analyses (e.g. total elemental analysis, P extractions, reactive transport modeling), we demonstrate that vivianite minerals began to precipitate below the lake-marine transition in the Bornholm Basin after the seawater intrusion. Vivianite authigenesis at depth resulted in a doubling of the total P concentrations in our sediments, thereby significantly altering the P burial record. With a sulfidization experiment, we confirm that vivianite rapidly dissolves under sulfidic conditions and show that its susceptibility to sulfide-induced dissolution does

not seem affected by the incorporation of Mn or Mg in the mineral.

## 2 Methods

### 2.1 Study site, coring and lithology

The Baltic Sea is a semi-enclosed basin in which the water column salinity is strongly affected by changes in sea level (Björck, 1995). The basin was a freshwater environment (the Ancylus lake) before the intrusion of North Sea water approximately 8900

5  years ago. This intrusion was followed by a gradual transition to more saline conditions and the establishment of the Littorina Sea (~8000 years ago; Andrén et al., 2011; Sohlenius et al., 2001). Large parts of the Baltic Sea experienced bottom water hypoxia ~7000-4000 years ago during the Holocene Thermal Maximum. This is assumed to be the result of the inflow of North Sea water and warmer sea surface temperatures, which together may have caused water column stratification and the subsequent development of widespread hypoxia in the Baltic Sea (Zillén et al., 2008).

The Bornholm Basin is located in the central Baltic Sea (Fig. 1), and is currently characterized by a surface water salinity of 7-8 and a deep water salinity of 16-20 (Mohrholz et al., 2006). At present, the bottom waters are seasonally hypoxic with minimum oxygen concentrations below 0.5 mg/L (Mohrholz et al., 2006). Sediment cores were recovered from three holes (A-C) at station M0065 in the Bornholm Basin during IODP Expedition 347 in October 2013 (55° 28.09' N, 15° 28.63' E; water depth: 84.3 m). We applied the mid-composite depth scale as reported in the proceedings of the expedition (mcd; Andrén et al., 2015). The upper two meters of sediment were not sampled due to the risk of mustard gas contamination. A

multicore was recovered from the same location in June 2016 during a research cruise with RV Pelagia for additional porewater analyses (0-50 cm sediment depth).

The lowest part of the long sediment record (below ~ 13.5 mcd) consists of greyish-brown to dark grey clays with a down-core increase in the silt and sand fraction. These sediments are overlain by grey clays of which the upper part is

dominated by iron sulfide laminations. The top sediments are composed of organic-rich clays containing fragments of bivalve shells and organic remnants (< 9.65 mcd, < 9.85 mcd and < 9.69 mcd for Holes A, B and C, respectively). These organic-rich clays contain benthic foraminifers and brackish-marine diatom species, whereas the deeper sediments are generally characterized by freshwater diatom taxa and the absence of benthic foraminifera. These shifts in lithology and biostratigraphy



mark the lake-marine transition between ~ 9 and 10 mcd. A detailed description of the lithology and biostratigraphy can be found in Andrén et al. (2015).

## 2.2 Porewater analysis

Porewater was collected and filtered during the IODP 347 expedition using rhizons or squeezers (0.2 µm pore size). Alkalinity was calculated by the method of Grasshoff et al. (1983) after titration with 0.01 M HCl. Dissolved inorganic carbon (DIC) was measured in samples poisoned with saturated $HgCl_2$ using an AS-C3 analyzer (Apollo SciTech) that consisted of an acidification and purging unit in combination with a $CO_2/H_2O$ analyzer (LICOR-7000). The conductivity method in Hall and Aller (1992) was used to determine ammonium concentrations ($NH_4^+$) in the porewaters and dissolved sulfide (denoted as $HS^-$

) was measured using the methylene blue method of Cline (1969). Porewaters were acidified and analyzed for Mg, P, Ca, Fe and Mn by inductively coupled plasma - optical emission spectrometry (ICP-OES), and are assumed to represent $Mg^{2+}$, $PO_4$, $Ca^{2+}$, $Fe^{2+}$ and $Mn^{2+}$, respectively. Sulfate ($SO_4^{2-}$) and chloride ($Cl^-$) concentrations were determined by a Metrohm 882 compact ion chromatograph at the University of Bremen. Salinity was calculated from $Cl^-$ concentrations (Dickson et al., 2007). To determine concentrations of porewater methane ($CH_4$), 5 $cm^3$ sediment samples were extruded in glass vials filled

with 1 M NaOH solution, sealed and stored upside down for further analysis. After headspace injection (250 µL), the $CH_4$ concentrations were determined using an Agilent A7890 gas chromatograph. More details on the porewater procedures can be found in Andrén et al. (2015). The methods used to analyze porewaters from the multicore are described in Egger et al. (2016).

## 2.3 Bulk sediment analysis

Sediment samples were collected at a 2 m depth resolution immediately after core retrieval and stored in nitrogen-purged aluminum bags to prevent sample oxidation (Hole C). Sediment samples were also collected at a 20 cm resolution during the on-shore sampling campaign (Hole A). The sediment record at and just below the lake-marine transition was studied in more

detail using samples collected in November 2014 at the Bremen Core Repository (5 cm resolution; Hole $A_{HR}$). All sediment

samples were freeze-dried and ground with an agate mortar and pestle at Utrecht University prior to further analysis.

Sediment subsamples (0.125 g) were dissolved overnight at 90 °C in a mixture of hydrofluoric acid, nitric acid and

perchloric acid. After evaporation of the acids at 140 °C, the residue was redissolved in a nitric acid solution (1 M). Total

5    concentrations of Mo, Fe, aluminum (Al), S, Mn, P, Ca and Mg in the solutions were determined by ICP-OES (Perkin Elmer

Optima 3000). The relative standard deviation for all elements was generally less than 3% (laboratory reference material (ISE-

921), in-house standards and sample replicates).

During the on-shore sampling party at the Bremen Core Repository, $C_{org}$ content was determined in decalcified

sediment samples with a LECO CS-300 carbon-sulfur analyzer (Holes A-C, see Andrén et al. (2015) for more details).

## 2.4 Sequential Fe and S extractions

We performed a sequential Fe extraction on dry sediment subsamples (0.1 g) from Hole C according to the method of Poulton

and Canfield (2005) in an oxygen-free atmosphere. The extractants to target the various Fe phases, i.e. carbonate-associated

Fe (Fe-carb), easily reducible Fe-oxides (Fe-ox1), crystalline Fe-oxides (Fe-ox2) and magnetite (Fe-mag), are summarized in

15   Table 1. The Fe-ox1 and Fe-ox2 fractions together define the sedimentary Fe-oxide fraction in the sediment (Fe-ox; without

magnetite). The 1,10-phenanthroline method was used to determine Fe concentrations in the solutions (APHA, 2005).

For the same set of sediment samples we determined the acid volatile sulfide (AVS) and chromium-reducible sulfide

(CRS) fractions with the S extraction method as developed by Burton et al. (2008). We used a 6 M HCl/0.1 M ascorbic acid

solution and an acidic chromium(II) solution to convert sulfide in AVS and CRS into hydrogen sulfide gas. The gas was

20   trapped in an alkaline zinc-acetate solution, in which the amount of sulfide was then measured by iodometric titration (APHA,

2005). The sediment samples were shielded from the atmosphere to avoid sample oxidation, using either a glovebox (AVS) or

a nitrogen gas line (CRS). The AVS fraction is assumed to represent iron monosulfides (FeS) and the CRS fraction is assumed

to represent pyrite ($FeS_2$). The CRS concentrations (in µmol S/g) were divided by two to represent concentrations of $FeS_2$ (in

µmol FeS$_2$/g). The relative standard deviations were generally below 10% for both the Fe and S extraction, based on replicates and in-house standards.

## 2.5 Bulk phosphorus analysis (SEDEX and P XANES)

The P fractionation in ground sediment subsamples (0.1 g) was investigated with the SEDEX method (Ruttenberg, 1992), as modified by Slomp et al. (1996), with the inclusion of the exchangeable P step. The extraction steps are summarized in Table 2. The P concentrations in the citrate-dithionite-bicarbonate (CDB) solutions were measured by ICP-OES. The P concentrations in the other extractants were measured colorimetrically by the molybdenum blue method (Strickland and Parsons, 1972). For Holes C and A$_{HR}$, the total extracted P was almost similar to the total P concentrations as derived by total

destruction (< 15 % difference). For Hole A, the sum of all extracted elements for the sediments from Hole A slightly underestimated the total P content (on average 18 % lower). The relative standard deviations, as based on sample triplicates and in-house standards, were generally below 10 %. To avoid changes in P fractionation related to sample oxidation (Kraal et al., 2009; Kraal and Slomp, 2014), samples that were collected during the expedition and on-shore sampling (Holes C and A) were not exposed to oxygen during sample preparation and the first six steps of the P extraction. The samples from hole A$_{HR}$

were exposed to oxygen for over a year before sample collection and were therefore not treated anoxically during the SEDEX. Trends with sediment depth were, however, similar between the different holes and sampling campaigns, suggesting no distinct changes in P fractionation due to sample oxidation.

Apart from the sequential P extractions, we investigated the P fractionation in a sediment sample from 11.67 mcd (Hole C) with unfocused X-ray absorption near edge structure (XANES) spectroscopy. The analysis was conducted in April

2015 at the ID21 beamline at the European Synchrotron Radiation Facility (ESRF) in Grenoble, France (see Salomé et al. (2013) for details on the beamline). Prior to analysis, the monochromator was calibrated against the first derivative of tricalcium phosphate (2149.43 eV). The ground sediment was fixed between Kapton® tape and analyzed under vacuum with an unfocused beam in fluorescence mode at the P K-edge (2130 - 2300 eV; room temperature). Background removal and normalization were performed using the ATHENA software package (Ravel and Newville, 2005). We applied a 3-point



smoothing on the mean spectrum (# of individual spectra = 35) to reduce overall noise. All standards are described in Dijkstra et al. (2016).

## 2.6 SEM-EDS of blue spherical aggregates

While examining sediments with a light microscope, we observed blue spherical aggregates in sediments from 11.98 mcd (Hole $A_{HR}$). These aggregates were coated with carbon, mounted on an aluminum stub with carbon tape and investigated by scanning electron microscope energy dispersive spectroscopy (SEM-EDS; JCM 6000PLUS NeoScope Benchtop SEM). The imaging was conducted in backscatter mode with an acceleration voltage of 15 kV. We performed EDS in the 0-20 keV energy range for elemental quantification of the aggregates (probe current: 1 nA; acquisition time: 50 s). The SEM-EDS software was

used to estimate the relative abundances in mol% for the major elements (Standardless ZAF procedure). Elements associated with salt particles and clays (i.e. Na and K) were on average below 2 mol% and were not included in the results.

## 2.7 Vivianite sulfidization experiment

We performed a sulfidization experiment to examine the stability of vivianite in the presence of sulfide. Various P-bearing

minerals were synthesized by mixing a 0.2 M Mohr salt solution ($(NH_4)_2Fe(SO_4)_2 \cdot 6H_2O$) with a solution of 0.16 M ammonium acetate and 0.47 M disodium phosphate in a glovebox (1:1 volume ratio) following procedures described by Rouzies et al. (1993). We also added 0.6 M $MgCl_2$ and 0.6 M $MnCl_2$ solutions in various ratios to the Fe-P solution to synthesize minerals with varying Fe, Mg and Mn concentrations. All chemical solutions were deoxygenated before mixing and prepared in sulfate-free artificial seawater with a salinity of 14, which is equal to the porewater salinity at the P peak location in the sediments

(Fig. 3). After 24 hours, the precipitates were filtered and washed twice with deoxygenated water and dried in the glovebox for 7 days. A small subsample (0.01-0.1 g) of each precipitate was destructed in hydrofluoric acid, nitric acid and perchloric acid for 24 hours. After evaporation of the acids, the residue was redissolved in 1 M nitric acid and analyzed by ICP-OES for P, Fe, Mn and Mg concentrations (relative standard deviation was 11 % on average; based on duplicates and triplicates). The mineralogy of the precipitates was determined by XRD using a Bruker D2 diffractometer with Cobalt Kα radiation over a 5-



85 2θ range with a step size of 0.026°. Two precipitates (1 and 4) were also examined by SEM-EDS (see sect. 2.6 for technical details).

We synthesized three vivianite minerals (precipitate 1-3) and two mixtures of vivianite and struvite ($MgNH_4PO_4.6H_2O$; precipitate 4 & 5) (Table 3). The (Fe+Mg+Mn)/P ratio for the vivianite precipitates is between 1.44-1.49 mol/mol (Table 3), and thus close to the theoretical stoichiometric ratio of 1.5 mol/mol (Nriagu, 1972). The total fraction of Mg, Mn, Fe relative to total P ($Mg_{tot}$, $Mn_{tot}$, $Fe_{tot}$; Table 3) can be used to estimate the relative contribution of vivianite and struvite to the mixed precipitates at the start of the experiment ($P_{stru}$ and $P_{viv}$). The elemental fractions in vivianite and struvite are expressed as:

$$\frac{(Mg_{viv}+Mn_{viv}+Fe_{viv})}{P_{viv}} = 1.5 \qquad (1)$$

$$\frac{Mg_{stru}}{P_{stru}} = 1.0 \qquad (2)$$

Equations (1-2) can be solved with the following assumptions: 1) Fe and Mn are only incorporated in vivianite (i.e. $Mn_{tot} = Mn_{viv}$ and $Fe_{tot} = Fe_{viv}$) and 2) the fraction of Mg in vivianite is 14% of the P fraction in vivianite, i.e. $Mg_{viv} = 0.14\ P_{viv}$, in accordance with the Mg/P-ratio of vivianite precipitates 2 and 3 (Table 3). Rearranging terms leads to Eq. (3) with one unknown parameter ($P_{viv}$);

$$\frac{0.14P_{viv}+Mn_{tot}+Fe_{tot}}{P_{viv}} = 1.5\frac{Mg_{tot}-0.14P_{viv}}{1-P_{viv}} \qquad (3)$$

Using Eq. (3), we estimated that 80 % of P in precipitate 4 is incorporated in vivianite and 20 % in struvite ($P_{stru} = 1-P_{viv}$). Precipitate 5 is estimated to contain more struvite than precipitate 4, with 42 % of all P being present in struvite.

Small subsamples of the five precipitates (~ 5 mg) were inserted into 50 ml centrifuge tubes that contained the artificial seawater solution (salinity of 14) with 0 or 1000 µM sulfide (30-50 ml seawater, depending on the mass of the subsamples). The additional weight of the eppendorf tubes due to replacement of air by argon in the argon-filled glovebox was subtracted from the sample masses. All experiments were performed in duplicate (2x2x5 samples in total). The samples were stored in aluminum bags filled with $N_2$ gas and placed on a shaker. At eight time steps within 408 hours, the samples were centrifuged and the supernatant was sampled in the glovebox for sulfide measurements (200 µL in a 4.5 % Zn acetate solution)




and P analysis (1 ml to 20 µL 0.2 M HCl). The sulfide concentrations in the zinc-acetate solutions were measured colorimetrically with a diamine solution (FeCl$_3$ and N,N-dimethyl-p-phenylenediamine sulfate) according to Cline (1969). The molybdenum blue method of Strickland and Parsons (1972) was used to measure P concentrations. The P concentrations were multiplied by the solution volume (corrected for volume loss due to sampling), and divided by the sample mass to calculate

relative percentages of dissolved P. For three samples, the measured sulfide concentrations in tubes with precipitate 3 were much lower than observed for all other vivianite samples from the same time steps (< 35 µM at 0.5, 4 and 24 hours). The low concentrations were attributed to a measurement error and excluded in Fig. 9.

### 2.8 Reactive transport model

### 2.8.1. General description

We developed a 1-dimensional reactive transport model to investigate the temporal evolution of vivianite in the Bornholm Basin and its sensitivity to changes in salinity, primary productivity and bottom water redox conditions. The model describes the temporal evolution of 27 species (Table 4) via a set of mass conservation equations that include physical transport as well as biogeochemical transformations. Their generic form for solids and solutes is, respectively,

$$\frac{\partial C_S}{\partial t} = \frac{1}{(1-\phi)}\frac{\partial}{\partial x}\left(-v(1-\phi)C_s\right) + \sum R_S \tag{4}$$

$$\frac{\partial C_p}{\partial t} = \frac{1}{\phi}\frac{\partial}{\partial x}\left(D'\phi\frac{\partial C_p}{\partial x} - u\phi C_p\right) + \sum R_p \tag{5}$$

where $C_S$ is the concentration of the solid species (mol cm$^{-3}$; mass per unit volume of solids), $C_p$ is the concentration of the solute (mol cm$^{-3}$; mass per unit volume of porewater), t is time (y), $\phi$ is sediment porosity, x is distance from the sediment-water interface (cm), $D'$ is the respective molecular diffusion coefficient in the sediment (D$_m$; cm$^2$ y$^{-1}$) at *in situ* conditions and corrected for tortuosity ($\theta^2$; Table S2; Boudreau, 1996), v and u are the advective velocities (cm y$^{-1}$) of the solid phase and

porewater, respectively, and $\sum R_S$ and $\sum R_p$ are the net reaction rates (mol cm$^{-3}$ y$^{-1}$) per unit volume of solids and porewater, respectively. Note that bioturbation and biodiffusion are not included in Eq. (4-5) since the Bornholm Basin is assumed to have been hypoxic during the main period of interest (7500 BP – present), thus implying limited benthic faunal activity. Porosity was described by an exponentially decreasing function of the form



$$\phi(x) = \phi_{\infty}(\phi_0 - \phi_{\infty})\exp(-\frac{x}{\gamma}) \qquad\qquad (6)$$

The parameters $\phi_{inf}$ (porosity at depth), $\phi_0$ (porosity at the sediment-water interface) and $\gamma$ (porosity e-folding distance, cm) were fit to the observed porosity profile (Fig. S1; Table S3), which was assumed to be constant over the simulation time. As a result, we calculated advective velocities assuming steady-state compaction (Boudreau, 1997; Meysman et al., 2005).

5    Biogeochemical reactions and their corresponding rate laws are given in Tables 5 and 6, respectively. Rate constants and other reaction parameters were mostly taken from the literature or, in case no fit to the data could be obtained, constrained using the model (Table 7). Both Fe-oxides and organic matter were divided into highly reactive ($\alpha$), less reactive ($\beta$) and inert ($\gamma$) phases. In comparison with $Fe(OH)_3^{\alpha}$, the $\beta$ phase is not used by organoclastic Fe reduction. This assumption is made to ensure that part of the reactive Fe-oxides is buried even under higher organic matter loading. The phosphate minerals apatite

10 ($Ca_5(PO_4)_3OH$) and vivianite were included in the model as sinks of porewater $PO_4^{3-}$. Their formation was described by second-order kinetics (Egger et al., 2016; Rooze et al., 2016). In line with Egger et al. (2016), vivianite was subject to dissolution under sulfidic (euxinic) porewater conditions. Siderite ($FeCO_3$) formation and dissolution was described similarly to vivianite kinetics. The model additionally included linear adsorption of $NH_4^+$ (Mackin and Aller, 1984; Soetaert and Herman, 2009).

    Boundary conditions at the sediment-water interface were prescribed as concentrations for the solutes and fluxes for

15 the solids (Table S4). We applied a zero-gradient boundary condition to all species at the lower boundary. The model code was written in R (version 3.4.0) using the *marelac* package (Soetaert et al., 2010a) for chemical and physical constants and the calculation of diffusion coefficients. Physical transport was calculated with the *ReacTran* package (Soetaert and Meysman, 2012). The depth domain, which describes the upper 50 m of the sediment, was discretized into 450 layers with highest resolution at the top and between 9-12 m. The set of equations obtained upon discretization of the mass conservation equations

20 was solved with the lsoda ordinary differential equation solver (Hindmarsh, 1983; Petzold, 1983) that is included in the *deSolve* package (Soetaert et al., 2010b).

### 2.8.2. Reconstructing the diagenetic history of the Bornholm Basin

The model was used to describe the diagenetic history of the Bornholm Basin over the past 25000 years, thereby assuming no changes in environmental conditions or fluxes before 20000 years BP. A spin-up period of 55000 years was used to ensure no effect of the initial conditions on the model solution in the major zone of interest, i.e. around the lake-marine transition.

Sedimentation rate for marine phase was estimated at 1.6 mm/y and kept constant for the entire simulation period, as we have no information on the sedimentation rate in the lake phase. This rate corresponds to a lake-marine transition at ~7.5 kyr as estimated by Zillen et al. (2008).

The flux of organic matter (OM) entering the sediment was fit to the observed $C_{org}$ profile, thereby assuming that in the marine phase the ratio of the various organic matter fractions obeyed the multi-G model as determined for fresh planktonic material,

i.e. 50% highly reactive, 16 % less reactive and 34 % refractory organic matter (Westrich and Berner, 1984; Fig. 2). Similarly, we varied bottom-water salinity conditions such that a best fit to the observed porewater Cl$^-$ profile was obtained. Iron oxides enter the sediment in a temporally varying ratio of its most reactive ($Fe(OH)_3^{\alpha}$), less reactive ($Fe(OH)_3^{\beta}$) and crystalline ($Fe(OH)_3^{\gamma}$) forms. We assumed that all Fe-oxides deposited before 20000 years BP were crystalline. The Fe loading was furthermore adjusted to fit the observed sedimentary profiles of Fe-oxide and S. During the onset of the Holocene Thermal

Maximum (7500-4500 years BP), an interval of high productivity and widespread anoxia, we assumed that part of the Fe-oxides had been transformed to pyrite before settling onto the sediment surface. After the termination of bottom water euxinia, which we set at 7000 BP, bottom waters were assumed to remain anoxic. Even though periods of oxygenation have occurred since then (Jensen et al., 2017; Zillén et al., 2008), the average bottom water oxygen concentration was likely low.

## 3 Results

### 3.1 Porewater data

Salinity and Mg$^{2+}$ concentrations are high in the brackish-marine sediments and gradually decrease with sediment depth (Fig. 3). The brackish-marine deposits are also characterized by high porewater alkalinity and DIC, as well as high NH$_4^+$ and PO$_4$

concentrations, which are all key products of organic matter decomposition. While alkalinity, DIC and $NH_4^+$ gradually decrease with sediment depth, a sharper decline across the lake-marine transition is observed for $PO_4$. The porewaters below 13 mcd contain less than 0.15 mM $PO_4$. In all sediments, porewater $Ca^{2+}$ concentrations are above 5 mM. Porewater $Fe^{2+}$ concentrations vary strongly with low values in the brackish-marine sediments ($< 0.1$ mM) and high values in the Ancylus lake sediments just

5 below the lake-marine transition ($> 0.8$ mM). The porewaters in sediments around the lake-marine transition are high in $Mn^{2+}$ ($> 0.2$ mM), and porewater sulfide is only detected in the upper 5 m of the sediment ($< 3$ mM). Porewater $SO_4^{2-}$ is high at the sediment-water interface (14 mM) and sharply declines with sediment depth to values below 1 mM in the deeper sediments ($< 2$ m sediment depth). Dissolved $CH_4$ shows high concentrations of up $> 11$ mM in the porewaters of the brackish-marine sediments, and decreases to $< 2$ mM below 30 mcd.

**3.2 Organic carbon and total elemental concentrations**

The brackish-marine sediments are more organic-rich than the deeper lake sediments ($> 3$ wt% versus $\sim 0.5$ wt%; Fig. 4). Sedimentary Mo, Mn and S concentrations are highest at the lake-marine transition (0.6, 356 and 1000 µmol/g, respectively), and generally elevated in the brackish-marine sediments when compared to the Ancylus lake deposits. Note that the lake sediments at or just below the lake-marine transition have elevated total S contents and sedimentary Fe/Al ratios above 0.7

15 wt%/wt%. We further observe a minimum in sedimentary Al, Mg and P concentrations between 9.5-11.5 mcd, and a P enrichment of $\sim 60$ µmol/g at 12 mcd. Total Ca is elevated in the lake sediments. The trend in $C_{org}$/P-ratios generally follows the $C_{org}$ trend with highest $C_{org}$/P-ratios ($> 75$ mol/mol) in the brackish-marine deposits.

**3.3 Sulfur, iron and phosphorus speciation (including P XANES)**

The brackish-marine deposits contain on average 3 µmol $FeS_2$/g and 200 µmol $FeS_2$/g whereas the lake sediments below 11 mcd are low in both FeS and $FeS_2$ ($< 1$ and $< 25$ µmol/g, respectively; Fig. 5). The highest concentrations of FeS and $FeS_2$ are observed just below the lake-marine transition at 10.3 mcd (10 and 300 µmol/g, respectively). The brackish-marine sediments

generally contain less Fe-carbonates and Fe-oxides than the lake sediments. Magnetite is a minor Fe phase throughout the record (< 50 µmol/g).

Exchangeable P is only a minor P fraction in the Bornholm basin sediments (< 5 % of total P at each sediment depth; Fig. 6). Organic P is highest in the brackish-marine deposits and decreases to values of < 3 µmol/g at depth in the lake

sediments. Ca-P is a major P phase throughout the sediment record, with lowest values in sediments just below the lake-marine transition. There are distinct peaks in sedimentary Fe-bound P, and to a lesser degree in exchangeable P, between 11.5 and 12 mcd. Maxima in Fe-bound P are located at 11.5 mcd for Hole A, 11.67 mcd for Hole C and 11.90 mcd for Hole $A_{HR}$. At these depths, Fe-bound P accounts for up to 63 % of the total sedimentary P pool.

Bulk P XANES spectroscopy was performed on sediments from 11.67 mcd (Hole C) for a more detailed investigation

of the sedimentary P fractionation (Fig. 7). The results do not provide evidence for the presence of vivianite in the bulk sediments as the characteristic post-edge oscillations of vivianite are not observed in the sediment sample. In addition, the position of the main peak (i.e. white line) of the bulk sediment is slightly shifted to higher energies compared to the white line of vivianite (difference: 0.1-0.2 eV). The spectrum of the sediment sample also does not resemble the almost featureless spectra of rhodochrosite-P and hureaulite either. Instead, the P XANES spectrum is most comparable to the P XANES spectrum of

fluorapatite with a shoulder feature and oscillations at similar energies (2163.5 and 2170 eV). Linear combination fitting of the bulk sediments was not possible due to the high amplitude of the white line caused by a strong self-adsorption effect.

## 3.4 Examination of blue aggregates with SEM-EDS

The blue spherical aggregates from 11.98 mcd (Hole $A_{HR}$) are approximately 400 µm in size (Fig. 8). Elemental quantification

of the aggregate surfaces with SEM-EDS suggests that the aggregates are enriched in Fe (between 13 to 83 mol%). Other major elements are P (0.2-17.4 mol%) and Mn (0.8-10.2 mol%). We also detected some Mg on the surfaces of both aggregates (0.8-8.2 mol%). The enrichments in Si and Al likely reflect clay particles that are attached to the aggregates.

**3.5 Sulfidization experiment**

Within 17 days (408 hours), most of the three vivianite precipitates had dissolved in the seawater solution that initially contained 1000 μM sulfide (> 75 %; Fig. 9B). Almost no vivianite was dissolved in the sulfide-free seawater (generally < 10 % of all P). The mixed vivianite/struvite precipitates show a different trend with time compared to the pure vivianite minerals (Fig. 9C). Only 25 % of all P from precipitate 4 was dissolved in the sulfide-rich seawater solution. The percentage of dissolved P from precipitate 5 varied widely among the duplicate samples resulting in large standard deviations (error bars). In contrast to the pure vivianite samples, also some P from the vivianite/struvite precipitates dissolved in the seawater solutions that did not contain sulfide. The gradual decrease in sulfide contents with time in all samples (Fig. 9D and 9E) is assumed to be the result of the escape of sulfide gas from the solutions during sampling.

**3.6 Model fits and transient evolution of solutes and particulates**

The reactive transport model reproduces the trends in all key porewater and solid phase profiles, including the observed enrichment in P at ~11.5 mcd (Fig. 3-6). The vast majority of P in this peak is present as vivianite (Fig. 6). The time-depth profiles of DIC and $C_{org}/P_{tot}$ confirm that their temporal evolutions mainly result from the imposed increase in organic matter loading (Fig. 11). The pattern of $PO_4$ does not follow that of DIC and indicates removal processes acting both in the surface layer as well as deeper in the sediment through the whole marine phase. The enhanced Fe loading concurring with the transition from freshwater to brackish-marine conditions induced a high initial release of $Fe^{2+}$ into the porewater, leading to concentrations exceeding 6 mM. With no sulfide in the sediment and sufficient $PO_4$ and $Fe^{2+}$, conditions for favorable for vivianite authigenesis and a distinct layer of this mineral formed. However, the continuous supply of $SO_4^{2-}$ from the saline bottom water to the sediment and resulting sulfide production consumed the majority of the $Fe^{2+}$ such that it became depleted in the upper sediment around 6000 BP and sulfide started to accumulate. No vivianite was formed in this sulfide-bearing zone and any vivianite present dissolved, as indicated by the temporal evolution of the rates of both processes (VF = vivianite formation; VD = vivianite dissolution). Dissolved $Fe^{2+}$ was however still present at greater depths in the Ancylus lake

sediments and vivianite authigenesis continued at depth, limited by the supply of $PO_4$ through downward diffusion. This formation is still ongoing and its current depth-integrated rate is estimated to be 0.0024 mol P $m^{-2}$ $y^{-1}$.

Between 5000-2000 BP, an enhanced Fe loading increased consumption of sulfide until the point where dissolved $Fe^{2+}$ could build up again in the upper meter of the sediment, leading to a second zone of vivianite formation and a slowdown

of vivianite dissolution below this ferruginous zone. With the drop in Fe loading at 2000 BP and the resulting buildup of sulfide, this surficial vivianite dissolved again, such that no remnants of it are currently visible in the sedimentary record.

## 4 Discussion

### 4.1 The lake-marine transition in the Bornholm Basin

Intrusion of North Sea water transformed the Ancylus lake into the brackish-marine Littorina Sea ~ 8000 years ago (Andrén

et al., 2011). This is in line with the trends in porewater salinity and $Mg^{2+}$, a major constituent of seawater, which both indicate more marine conditions in the Bornholm Basin nowadays than during the early Holocene (Fig. 3). The lake-marine transition, as recorded in the biostratigraphy and lithology of our sediments between 9 - 10 mcd (sect. 2.1), led to the deposition of $C_{org}$-rich sediments (Fig. 4). The decomposition of part of the organic matter resulted in the release of alkalinity, DIC, $NH_4^+$ and $PO_4$ (Fig. 3) to the porewater. The slight increase in $C_{org}$ before the lake-marine transition may be caused by enhanced water

column productivity or by a lowering of the sedimentation rate due to decreased inputs of particulate matter in meltwater at that time, as proposed by Sohlenius et al. (1996). Such a shift in material supply may also explain the low concentrations of other terrestrial elements, such as Fe and Mg, between 10 and 11.5 mcd (Fig. 4) and the concurrent minimum in Ca-P (Fig. 6).

The Mo enrichments in the brackish-marine sediments at 9.5 mcd (> 0.3 µmol/g; Fig. 4) are indicative of seasonally hypoxic and possibly euxinic bottom waters (Scott and Lyons, 2012). This period of low oxygen conditions coincided with

the start of the Holocene Thermal Maximum just after the seawater intrusion when large parts of the Baltic Sea were experiencing bottom water hypoxia (Jilbert et al., 2015; Zillén et al., 2008). In the Bornholm Basin, periodic inflows of oxygenated North Sea water at that time may have resulted in short periods of bottom water oxygenation and, subsequently, Mn-carbonates may have formed in the surface sediments from Mn-oxides that precipitated onto the seafloor (Huckriede and



Meischner, 1996; Lenz et al., 2015). Some Mo could also have been transported to the sediments adsorbed to these Mn-oxides (Adelson et al., 2001; Algeo and Lyons, 2006), explaining our concurrent peak in Mo and Mn at 9.5 mcd (Fig. 4). The sediments at 9.5 mcd are also enriched in Fe and S due to the enhanced burial of Fe-sulfides (Fig. 4 & 5). Furthermore, sediments elevated in Fe and S (as Fe-sulfides) just below the lake-marine transition (Fig. 4 & 5) indicate that porewater sulfide

diffused into the deeper lake sediments during the early Littorina Sea stage. Such sulfidization fronts are widely observed in sedimentary records of the Pleistocene and Holocene Baltic Sea (Boesen and Postma, 1988; Böttcher and Lepland, 2000; Holmkvist et al., 2014; Sohlenius et al., 1996).

## 4.2 Post-depositional vivianite authigenesis below the lake-marine transition

The distinct enrichments in P below the sulfidization front at ~11.5 mcd (Fig. 4) can be attributed to the presence of vivianite in the sediments. Vivianite dissolves in the CDB-extraction step that targets Fe-bound P (Dijkstra et al., 2016; Nembrini et al., 1983), and may therefore be responsible for the enrichments in both Fe-P and total P (Fig. 4 & 6). Vivianite is also extracted in the hydroxylamine-HCl step of the sequential Fe-extraction and may thus explain our peak in sedimentary Fe-oxides at 11.67 mcd (Fe-ox; Fig. 5). In addition, we discovered Fe-P aggregates in the P-rich sediments (Fig. 8) that were almost identical

in shape, color and size as vivianite minerals synthesized by Zelibor et al. (1988). Our aggregates contained Mg and Mn (Fig. 8), which are impurities that have been observed in many vivianite minerals in marine and lake deposits (e.g. Dijkstra et al., 2016; Egger et al., 2015a; Hsu et al., 2014). Finally, the reactive transport model also calculates a vivianite peak concurring with the observed enrichments in P at ~11.5 mcd, while at the same time reproducing the trends in all key porewater and solid phase profiles (Fig. 3-6 & 11). Our sequential extractions, SEM-EDS and model analyses thus clearly point towards the

presence of vivianite in the P-rich sediments of the Bornholm Basin.

     The presence of vivianite was, however, not confirmed by synchrotron-based XAS, as the bulk sediment P XANES spectrum (11.67 mcd; Hole C) did not resemble the spectrum of our vivianite standard (Fig. 7). Instead, the post-edge oscillations of the sediment P-XANES spectrum were typical for apatites (Fig. 7), even though the sediment is assumed to contain twice as much Fe-bound P compared to authigenic Ca-P (such as fluorapatite) (Fig. 6). This discrepancy between

synchrotron-based and SEDEX-derived P fractionation might be caused by a bias of P XANES spectroscopy towards P species at particle surfaces, as has been proposed earlier by Dijkstra et al. (2016). As a consequence, the P XANES spectrum might mainly reflect a more disperse (apatite) pool instead of individual vivianite aggregates. This is line with the study of Egger et al. (2015a), in which P XANES spectroscopy was unable to conclusively identify vivianite in vivianite-rich surface sediments

in the Bothnian Sea.

### 4.3 Formation and stability of post-depositional vivianite

In the Ancylus lake sediments, vivianite authigenesis occurs at the interface where downward diffusing $PO_4$, which is released by organic matter decomposition in the brackish-marine deposits, meets upward diffusing $Fe^{2+}$ (Fig. 3). The high $Fe^{2+}$

concentrations in the Ancylus lake sediments are assumed to be the result of reductive dissolution of Fe-oxides in the lake sediments (Fig. 5) via organoclastic Fe-reduction, Fe-mediated anaerobic oxidation of $CH_4$ or a combination of both (Egger et al., 2017). Although porewater $PO_4$ already began to accumulate in the surface sediments before the lake-marine transition, following the increasing in organic matter inputs at 10500 BP, the rate of vivianite formation resulting from this is insignificant compared to the authigenesis taking place directly after the lake-marine transition, explaining the fast increase in vivianite

concentration at the interface with the lake sediments (Fig 11).

      A key variable in the temporal evolution of vivianite is the presence of free sulfide. Although it is currently not visible in the sedimentary record, the temporary second vivianite formation front active between 5000-2000 BP (Fig. 11) had an important implication: it led to a disconnection of the major vivianite front from the active sulfidization front. The downward diffusion of sulfide shortly after the lake marine-transition promoted dissolution of freshly precipitated sedimentary vivianite,

resulting in a gradual migration of the vivianite peak into deeper sediments below the lake-marine transition. This process was hindered by the increased Fe loading and resulting enhanced sulfide consumption, removing the tight connection between the sulfidization and vivianite fronts and decelerating the downward migration of the vivianite front over time. At present, pore water sulfide is restricted to the upper 5 m of sediment (Fig. 3), suggesting that the present vivianite peak is stable with respect to its position relative to the lake-marine transition (now at 11.5 -12 mcd; Fig. 6). The location of the vivianite peak in the

sediments thus depends on the susceptibility of the vivianite to dissolution by dissolved sulfide and the extent of downward



diffusion of sulfide. This key role for sulfide is also highlighted in our sulfidization experiment in which vivianite dissolves within 100 hours in a seawater solution containing sulfide, also when it contains Mn or Mg (Fig. 9).

Free sulfide also determines the height and thickness of the vivianite peak, as is shown by a model run in which we assumed that no salinization took place (i.e., salinity was kept at its initial value for the entire model simulation time; Fig. 12).

Under such conditions, a broad vivianite peak developed with a maximum at ~6 mcd and a depth-integrated concentration twice as high as in the default model run, as no sulfide-induced dissolution of vivianite took place. The stability of the vivianite enrichment relative to the lake-marine transition is confirmed by running the model for an additional 3000 years without any changes in environmental conditions (Fig. 12). Over the course of this simulation, the continuous supplies of $PO_4$ from above and $Fe^{2+}$ from below the front led to an increase in the depth-integrated vivianite concentration from 20.4 to 25.2 mol P m$^{-2}$.

A final model sensitivity analysis emphasizes the key role of enhanced productivity and associated organic matter deposition on vivianite formation. If the organic matter loading is kept at its initial value for the last 10500 years, only a very small amount of vivianite forms at the lake-marine transition (Fig. 12).

Note that we synthesized struvite together with vivianite in our sulfidization experiment (precipitate 4 & 5; Table 3; Fig. 9-10). In contrast to vivianite, struvite is observed in sulfide-rich waste water (e.g. Charles et al., 2006) and may thus be

stable in the presence of sulfide. This is confirmed by our sulfidization experiment, in which, for example, only 25% of precipitate 4 (estimated struvite fraction: 20 mol% P) dissolved in sulfide-rich seawater (Fig. 9). Struvite is observed in freshwater sediments (Cohen and Ribbe, 1966; Donovan and Grimm, 2007; Pi et al., 2010), and porewater saturation calculations indicate that struvite might also be present in some brackish-marine systems (e.g. Kau Bay and Saanich inlet; Middelburg, 1990; Murray et al., 1978). In order for struvite to precipitate, porewaters need to be elevated in $NH_4^+$ and $PO_4$

and have high molar Mg/Ca porewater ratios (e.g. above 580 mol mol$^{-1}$ by pH 7.6-7.9 and a $PO_4$ concentration of 10 μM; Gull and Pasek, 2013), conditions that are not met in the Bornholm Basin (Fig. 3). So although struvite precipitation may affect P burial in other settings, it likely does not affect vivianite authigenesis in sediments of the Bornholm Basin.



### 4.4 Implications for sedimentary P records

The presence of post-depositional vivianite in sediments below the lake-marine transition is a widespread phenomenon in the Baltic Sea (Fig. 13). Vivianite has been observed below the lake-marine transition in the Landsort Deep (Dijkstra et al., 2016), in the Gulf of Finland (Winterhalter, 1992) and in the Southern Baltic Sea (Sviridov and Emelyanov, 2000). Vivianite is also

likely present in Ancylus lake sediments in the coastal Bothnian Sea (Dijkstra et al., 2017) and may explain the enrichments in CDB-extractable P (Fe-bound P) in lake sediments from the Archipelago Sea in the Northern Baltic Sea (Virtasalo and Kotilainen, 2008) and the large cm-scale blue aggregates observed in Ancylus lake sediments in the Little Belt, Denmark (N. van Helmond, personal communication).

Vivianite may also have precipitated at depth in sediments of other basins that have experienced past variations in

salinity and productivity. For instance, large spherical vivianite aggregates have been found in Laptev Sea sediments (Taldenkova et al., 2010). These crystals were detected below sediments that were deposited during a period of enhanced water column primary productivity (as reflected by abundant ostracods; Stepanova et al., 2012). Vivianite in the Laptev Sea may thus have formed post-depositionally as a result of a change in primary productivity.

In our study, we clearly demonstrate that post-depositional vivianite formation can result in sediment P enrichments

that do not reflect changes in primary productivity and P burial at the time of sediment deposition. In the Bornholm Basin, this has led to a near doubling of the total P concentration in the sediment. Post-depositional P mineral formation can also result in a distinct lowering of the $C_{org}$:P-ratio. In our sediments, the increase in P coincides with an increase in $C_{org}$, making this change less visible (Fig. 4). We conclude that the presence of post-depositional vivianite should be considered when interpreting sedimentary records of P in systems that are subject to environmental perturbations (e.g. shifts in primary productivity).

The long-term stability of post-depositional vivianite and its vertical location in sediments is controlled by the Fe, S and $CH_4$ dynamics in the sediments. This is illustrated by a simplified schematic of sediment and porewater profiles from the Bornholm Basin and from the sulfidic deep basin of the Black Sea (Fig. 14). A key difference between both systems is the location of the present sulfidization zone relative to the post-depositional vivianite peak in the sediment. In the Bornholm Basin, a sulfate/methane transition zone that is located above the lake-marine transition prohibits diffusion of sulfide into the

underlying lake sediments. Due to the subsequent lack of sulfide in the lake sediments, the present vivianite peak is stable with

respect to its position relative to the lake-marine transition and, under the same environmental conditions, will continue to grow at this location where downward diffusing $PO_4$ meets upward diffusing $Fe^{2+}$ (Fig. 12). The deep basin of the Black Sea is characterized by lower sedimentation rates and higher bottom water sulfide concentrations than the Bornholm Basin and, as a consequence, sulfide is diffusing into the lake sediments, promoting a downward migrating vivianite front. As has already

been discussed in detail by Egger et al. (2016), the vivianite front in the Black Sea has already migrated more than two meters below its initial location at the lake-marine transition.

## 5 Conclusions

In the Bornholm Basin, the presence of vivianite aggregates results in a distinct peak in total P (> 40 µmol/g) in the sediment

record. These P-bearing minerals form post-depositionally below the lake-marine transition in the sediments, where $PO_4$ from the marine organic-rich sediments meets dissolved $Fe^{2+}$ from the lake sediments. The vivianite front in the Bornholm Basin has migrated ~ 1.5 m downwards from its initial location at the lake-marine transition. At present, it is a stable feature in the lake sediments of the Bornholm Basin. Post-depositional vivianite may also have formed in other lake and marine basins that are subject to environmental perturbation, such as a change in primary productivity, which can be associated with a lake-

marine transition. The possibility of post-depositional vivianite formation should thus be considered when using sediment P proxies to reconstruct paleo-environmental conditions in coastal basins.

## 6 Data availability

The data files are available from the PANGAEA database (https://doi.pangaea.de/10.1594/PANGAEA.880135) or are already

published in Egger et al. (2017) or Andrén et al. (2015).

**Competing interests**

The authors declare that they have no conflict of interest.

**Acknowledgements**

This research was funded by the European Research Council under the European Community's Seventh Framework Programme (FP7/2007-2013)/ERC Starting Grant 278364 and the Netherlands Organisation for Scientific Research (NWO Vici Grant 865.13.005). This research used samples and/or data provided by the Integrated Ocean Drilling Program (IODP). We thank the captain, crew and scientists on board on the Greatship Manisha from 12th of September until the 1st of November 2013 and the scientists that were part of the onshore science party from 22 January to 20 February 2014. Also the captain, crew

and scientists on board of R/V Pelagia from 28th of May to 15th of June 2016 (cruise 64PE411) are thanked. We also acknowledge the European Synchrotron Facility for providing beamtime at ID21. We further thank C. Rivard for her assistance at the beamlines, and E. Smedberg for providing us with the Baltic Sea bathymetric map. We also thank Fieke Mulders for her assistance with the sulfidization experiment and SEDEX.

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



# Tables

**Table 1: Details on the Fe extraction method by Poulton and Canfield (2005): chemical solutions and targeted phases.**

| Steps | Extractant | Time | Target phase | Term |
|-------|-----------|------|-------------|------|
| I | 1 M sodium acetate brought to pH 4.5 with acetic acid | 24 h | Carbonate-associated Fe | **Fe-carb** |
| II | 1 M hydroxylamine-HCl in 25% v:v acetic acid | 48 h | Easily reducible Fe-oxides FeS[1] | **Fe-ox1**[2] |
| III | 50 g/l sodium dithionite solution buffered to pH 4.8 with 0.35 M acetic acid/0.2 M sodium citrate | 2 h | Crystalline Fe-oxides | **Fe-ox2**[2] |
| IV | 0.2 M ammonium oxalate/0.17 M oxalic acid | 2 h | Magnetite | **Fe-mag** |

5    [1]: Some of the extracted Fe in this solution may represent FeS (Egger et al., 2015a).

[2]: The Fe-ox1 and Fe-ox2 fractions together define the sedimentary Fe-oxide fraction in the sediment (Fe-ox; without magnetite).



**Table 2: Chemical solutions, targeted phases and terms of the SEDEX** (Ruttenberg, 1992)**, including the modifications by Slomp et al. (1996).**

| Steps | Extractant | Time | Target phase | Term |
|---|---|---|---|---|
| I | 1 M MgCl$_2$ (pH 8) | 0.5 h | Exchangeable P | **Ex-P** |
| II$_a$ | CDB solution (pH 7.6): 0.3 M Na$_3$ citrate; 25 g/L Na dithionite; 1 M NaHCO$_3$ | 8 h | P bound to Fe-oxides Fe(II)-phosphates[1] | **Fe-P** |
| *II$_b$* | *1 M MgCl$_2$ (pH 8)* | *0.5 h* | *Washing step* | |
| III$_a$ | 1 M Na acetate buffered to pH 4 with acetic acid | 6 h | Authigenic apatite | **Authi Ca-P[2]** |
| *III$_b$* | *1 M MgCl$_2$ (pH 8)* | *0.5 h* | *Washing step* | |
| IV | 1 M HCl | 24 h | Detrital apatite | **Detr-P[2]** |
| V | Combustion at 550 °C 1 M HCl | 2 h 24 h | Organic P | **Org-P** |

[1]: Fe(II)-phosphates as vivianite can also dissolved in the CDB solution, as shown by Dijkstra et al. (2016) and Nembrini et al. (1983).

5    [2:] Authigenic Ca-P and detrital P are assumed to both represent Ca-phosphates with different susceptibilities towards acid dissolution and are summed as Ca-P.





**Table 3: Key characteristics of the synthesized phosphate minerals with molar iron(Fe), magnesium(Mg) and manganese(Mn) fractions in the precipitates relative to phosphorus (P). The mineralogy was investigated by XRD (Fig. 9A).**

| # | Mineral phase | $(Fe_{tot}+Mg_{tot}+Mn_{tot})/P_{tot}$ | $Fe_{tot}$ | $Mg_{tot}$ | $Mn_{tot}$ | Color |
|---|---|---|---|---|---|---|
| 1 | Vivianite | 1.49 | 1.40 | 0.09 | 0.00 | Dark blue |
| 2 | Vivianite | 1.44 | 1.24 | 0.14 | 0.06 | Dark blue |
| 3 | Vivianite | 1.44 | 0.99 | 0.14 | 0.31 | Blue |
| 4 | Vivianite & struvite | 1.41 | 0.74 | 0.31 | 0.36 | Blue |
| 5 | Vivianite & struvite | 1.34 | 0.81 | 0.52 | 0.01 | Light blue |



**Table 4. Chemical species included in the diagenetic model.**

| Species | Notation | Type |
|---|---|---|
| Organic carbon[a] | $C_{org}^{\alpha,\beta,\gamma}$ | Solid |
| Iron oxides[a] | $Fe(OH)_3^{\alpha,\beta,\gamma}$ | Solid |
| Iron oxide-bound phosphorus | $Fe_{ox}P^{\alpha,\beta,\gamma}$ | Adsorbed and co-precipitated |
| Iron monosulfide | $FeS$ | Solid |
| Pyrite | $FeS_2$ | Solid |
| Siderite | $FeCO_3$ | Solid |
| Elemental sulfur | $S_0$ | Solid |
| Vivianite | $Fe_3(PO_4)_2$ | Solid |
| Organic phosphorus | $P_{org}$ | Solid |
| Calcium-bound phosphorus (both apatite and detrital) | $CaP$ | Solid |
| Chloride | $Cl^-$ | Solute |
| Calcium | $Ca^{2+}$ | Solute |
| Oxygen | $O_2$ | Solute |
| Sulfate | $SO_4^{2-}$ | Solute |
| Iron | $Fe^{2+}$ | Solute |
| Hydrogen sulfide[b] | $\sum H_2S$ | Solute |
| Methane | $CH_4$ | Solute |
| Total ammonia[b] | $\sum NH_4^+$ | Solute |
| Nitrate | $NO_3^-$ | Solute |
| Total phosphate[b] | $\sum PO_4^{3-}$ | Solute |
| Dissolved inorganic carbon[b] | $DIC$ | Solute |

[a] There are three types of species: reactive (α), less reactive (β) and refractory (γ)

[b] $\sum$ denotes that all species of an acid are included



**Table 5. Reaction pathways and associated stoichiometries implemented in the diagenetic model.**

| | |
|---|---|
| **Primary redox reactions*†** | |
| $OM^{\propto,\beta} + aO_2 \rightarrow aCO_2 + bNH_4^+ + cH_3PO_4 + aH_2O$ | R1 |
| $OM^{\propto,\beta} + \frac{4a}{5}NO_3^- + \frac{4a}{5}H^+ \rightarrow aCO_2 + bNH_4^+ + cH_3PO_4 + \frac{2a}{5}N_2 + \frac{7a}{5}H_2O$ | R2 |
| $OM^{\propto,\beta} + 4aFe(OH)_3^\propto + 4a\chi Fe_{ox}P + 12aH^+ \rightarrow aCO_2 + bNH_4^+ + (c + 4a\chi)H_3PO_4 + 4aFe^{2+} + 13aH_2O$ | R3 |
| $OM^{\propto,\beta} + \frac{a}{2}SO_4^{2-} + aH^+ \rightarrow aCO_2 + bNH_4^+ + cH_3PO_4 + \frac{a}{2}H_2S + aH_2O$ | R4 |
| $OM^{\propto,\beta} \rightarrow \frac{a}{2}CO_2 + bNH_4^+ + cH_3PO_4 + \frac{a}{2}CH_4$ | R5 |
| **Secondary redox and other reaction equations†** | |
| $2O_2 + NH_4^+ + 2HCO_3^- \rightarrow NO_3^- + 2CO_2 + 3H_2O$ | R6 |
| $O_2 + 4Fe^{2+} + 8HCO_3^- + 2H_2O + 4\chi H_2PO_4^- \rightarrow 4Fe(OH)_3^\propto + 4\chi Fe_{ox}P + 8CO_2$ | R7 |
| $2O_2 + FeS \rightarrow SO_4^{2-} + Fe^{2+}$ | R8 |
| $7O_2 + 2FeS_2 + 2H_2O \rightarrow 4SO_4^{2-} + 2Fe^{2+} + 4H^+$ | R9 |
| $2O_2 + H_2S + 2HCO_3^- \rightarrow SO_4^{2-} + 2CO_2 + 2H_2O$ | R10 |
| $2O_2 + CH_4 \rightarrow CO_2 + 2H_2O$ | R11 |
| $2Fe(OH)_3^{\propto,\beta} + 2\chi Fe_{ox}P + H_2S + 4CO_2 \rightarrow 2Fe^{2+} + 2\chi H_2PO_4^- + S_0 + 4HCO_3^- + 2H_2O$ | R12 |
| $Fe^{2+} + H_2S \rightarrow FeS + 2H^+$ | R13 |
| $FeS + H_2S \rightarrow FeS_2 + H_2$ | R14 |
| $4S_0 + 4H_2O \rightarrow 3H_2S + SO_4^{2-} + 2H^+$ | R15 |
| $FeS + S_0 \rightarrow FeS_2$ | R16 |
| $SO_4^{2-} + CH_4 + CO_2 \rightarrow 2HCO_3^- + H_2S$ | R17 |
| $CH_4 + 8Fe(OH)_3^{\propto,\beta} + 8\chi Fe_{ox}P + 15H^+ \rightarrow HCO_3^- + 8Fe^{2+} + 8\chi H_2PO_4^- + 21H_2O$ | R18 |
| $Fe(OH)_3^\propto + \chi Fe_{ox}P \rightarrow Fe(OH)_3^\beta + \chi H_2PO_4^-$ | R19 |
| $Fe(OH)_3^\beta + \chi Fe_{ox}P \rightarrow Fe(OH)_3^\gamma + \chi H_2PO_4^-$ | R20 |
| $Fe^{2+} + CO_3^{2-} \rightarrow FeCO_3$ | R21 |
| $3Fe^{2+} + 2H_2PO_4^- \rightarrow Fe_3(PO_4)_2 + 4H^+$ | R22 |
| $FeCO_3 + H_2S \rightarrow FeS + HCO_3^- + H^+$ | R23 |
| $Fe_3(PO_4)_2 + 3H_2S \rightarrow 3FeS + 2H_2PO_4^- + 2H^+$ | R24 |

\* Organic matter ($OM^{\propto,\beta}$) is of the form $(CH_2O)_a(NH_4^+)_b(H_3PO_4)_c$, with 'a' = 1, 'b' = 16/106 and 'c' = 1/106. † $\chi$ refers to the P:Fe ratio of $Fe(OH)_3^{\propto,\beta,\gamma}$.

$R6$ = nitrification; $R7$ = Fe(OH)₃ formation; $R8$ = FeS oxidation; $R9$ = FeS₂ oxidation; $R10$ = H₂S oxidation; $R11$ = aerobic CH₄ oxidation; $R12$ = Fe(OH)₃ reduction by H₂S; $R13$ = FeS formation; $R14$ = pyrite formation (H₂S pathway); $R15$ = S₀ disproportionation; $R16$ = pyrite formation (polysulfide pathway); $R17$ = SO₄-AOM; $R18$ = Fe-AOM; $R19$ = conversion (i.e. crystallization) from α to β phase; $R20$ = crystallization from β to γ phase; $R21$ = siderite precipitation; $R22$ = vivianite formation; $R23$ = siderite dissolution with H₂S; $R24$ = vivianite dissolution with H₂S




**Table 6. Reaction equations implemented in the model.**

**Primary redox reaction equations**

$$R_1 = k_{\propto,\beta} OM^{\propto,\beta} \left( \frac{[O_2]}{K_{O_2}+[O_2]} \right) \qquad (E1)$$

$$R_2 = k_{\propto,\beta} OM^{\propto,\beta} \left( \frac{[NO_3^-]}{K_{NO_3^-}+[NO_3^-]} \right) \left( \frac{K_{O_2}}{K_{O_2}+[O_2]} \right) \qquad (E2)$$

$$R_3 = k_{\propto,\beta} OM^{\propto,\beta} \left( \frac{[Fe(OH)_3^\alpha]}{K_{Fe(OH)_3^\alpha}+[Fe(OH)_3^\alpha]} \right) \left( \frac{K_{NO_3^-}}{K_{NO_3^-}+[NO_3^-]} \right) \left( \frac{K_{O_2}}{K_{O_2}+[O_2]} \right) \qquad (E3)$$

$$R_4 = \Psi k_{\propto,\beta} OM^{\propto,\beta} \left( \frac{[SO_4^{2-}]}{K_{SO_4^{2-}}+[SO_4^{2-}]} \right) \left( \frac{K_{Fe(OH)_3^\alpha}}{K_{Fe(OH)_3^\alpha}+[Fe(OH)_3^\alpha]} \right) \left( \frac{K_{NO_3^-}}{K_{NO_3^-}+[NO_3^-]} \right) \left( \frac{K_{O_2}}{K_{O_2}+[O_2]} \right) \qquad (E4)$$

$$R_5 = \Psi k_{\propto,\beta} OM^{\propto,\beta} \left( \frac{K_{SO_4^{2-}}}{K_{SO_4^{2-}}+[SO_4^{2-}]} \right) \left( \frac{K_{Fe(OH)_3^\alpha}}{K_{Fe(OH)_3^\alpha}+[Fe(OH)_3^\alpha]} \right) \left( \frac{K_{NO_3^-}}{K_{NO_3^-}+[NO_3^-]} \right) \left( \frac{K_{O_2}}{K_{O_2}+[O_2]} \right) \qquad (E5)$$

**Secondary redox and other reaction equations**

$$R_6 = k_1[O_2][\textstyle\sum NH_4^+] \qquad (E6)$$

$$R_7 = k_2[O_2][Fe^{2+}] \qquad (E7)$$

$$R_8 = k_3[O_2][FeS] \qquad (E8)$$

$$R_9 = k_4[O_2][FeS_2] \qquad (E9)$$

$$R_{10} = k_5[O_2][\textstyle\sum H_2S] \qquad (E10)$$

$$R_{11} = k_6[O_2][CH_4] \qquad (E11)$$

$$R_{12} = k_7[Fe(OH)_3^{\propto,\beta}][\textstyle\sum H_2S] \qquad (E12)$$

$$R_{13} = k_8[Fe^{2+}][\textstyle\sum H_2S] \qquad (E13)$$

$$R_{14} = k_9[FeS][\textstyle\sum H_2S] \qquad (E14)$$

$$R_{15} = k_{10}[S_0] \qquad (E15)$$

$$R_{16} = k_{11}[FeS][S_0] \qquad (E16)$$

$$R_{17} = k_{12}[SO_4^{2-}][CH_4] \qquad (E17)$$

$$R_{18} = k_{13}[Fe(OH)_3^{\propto,\beta}][CH_4] \qquad (E18)$$

$$R_{19} = k_{14}[Fe(OH)_3^{\propto}] \qquad (E19)$$

$$R_{20} = k_{15}[Fe(OH)_3^{\beta}] \qquad (E20)$$

$$R_{21} = k_{16}[Fe^{2+}][DIC] \qquad (E21)$$

$$R_{22} = k_{17}[Fe^{2+}][\textstyle\sum PO_4^{3-}] \qquad (E22)$$

$$R_{23} = k_{18}[FeCO_3][\textstyle\sum H_2S] \qquad (E23)$$

$$R_{24} = k_{19}[Fe_3(PO_4)_2][\textstyle\sum H_2S] \qquad (E24)$$



**Table 7. Reaction parameters used in the diagenetic model.**

| Parameter | Symbol | Value | Units | Source | Range given by source (if applicable, value given by source) |
|---|---|---|---|---|---|
| Decay constant for $C_{org}{}^{\alpha}$ | $k_{\alpha}$ | 0.4 | $yr^{-1}$ | a, b, c | 0.05-0.07[a], 1.62[a,b,c] |
| Decay constant for $C_{org}{}^{\beta}$ | $k_{\beta}$ | 0.02 | $yr^{-1}$ | a, b | 0.00070-0.0086[a,b] |
| Limiting concentration of $O_2$ | $K_{O2}$ | 20 | $\mu mol\ L^{-1}$ | d | 1-30 (20) |
| Limiting concentration of $NO_3^-$ | $K_{NO3^-}$ | 4 | $\mu mol\ L^{-1}$ | d | 4-80 (2) |
| Limiting concentration of $Fe(OH)_3$ | $K_{Fe(OH)3}$ | 65 | $\mu mol\ g^{-1}$ | d | 65-100 (65, 100) |
| Limiting concentration of $SO_4^{2-}$ | $K_{SO42-}$ | 1.6 | $mmol\ L^{-1}$ | d | 1.6 |
| $C_{org}$:$P_{org}$ acceleration factor for $SO_4^{2-}$ reduction and methanogenesis | $\eta$ | 10 | - | e | 30 |
| Attenuation factor for $E4$ and $E5$ | $\Psi$ | 0.00157 | - | b | 0.00157 |
| Rate constant for reaction $E6$ | $k_1$ | 10,000 | $mmol^{-1}\ L\ yr^{-1}$ | d | ≈10,000 (5,000) |
| Rate constant for reaction $E7$ | $k_2$ | 140,000 | $mmol^{-1}\ L\ yr^{-1}$ | d | 140,000 |
| Rate constant for reaction $E8$ | $k_3$ | 300 | $mmol^{-1}\ L\ yr^{-1}$ | d | 300 |
| Rate constant for reaction $E9$ | $k_4$ | 1 | $mmol^{-1}\ L\ yr^{-1}$ | d | 1 |
| Rate constant for reaction $E10$ | $k_5$ | 160 | $mmol^{-1}\ L\ yr^{-1}$ | d | ≥ 160 (160) |
| Rate constant for reaction $E11$ | $k_6$ | 10,000,000 | $mmol^{-1}\ L\ yr^{-1}$ | d | 10,000,000 |
| Rate constant for reaction $E12$ | $k_7$ | 0.4 | $mmol^{-1}\ L\ yr^{-1}$ | f | 0.4 |
| Rate constant for reaction $E13$ | $k_8$ | 1 | $mmol^{-1}\ L\ yr^{-1}$ | b, f | 100 |
| Rate constant for reaction $E14$ | $k_9$ | 0.000315 | $mmol^{-1}\ L\ yr^{-1}$ | g | 0.0003 |
| Rate constant for reaction $E15$ | $k_{10}$ | 3 | $yr^{-1}$ | h | 3 |
| Rate constant for reaction $E16$ | $k_{11}$ | 7 | $mmol^{-1}\ L\ yr^{-1}$ | b, f | 7 |
| Rate constant for reaction $E17$ | $k_{12}$ | 0.01 | $mmol^{-1}\ L\ yr^{-1}$ | e | 0.01 |
| Rate constant for reaction $E18$ | $k_{13}$ | $1.23\ 10^{-6}$ | $mmol^{-1}\ L\ yr^{-1}$ | g, i | $1.6\ 10^{-7}$ [g], $4.6\ 10^{-7}$ [i] |
| Rate constant for reaction $E19$ | $k_{14}$ | 0.6 | $yr^{-1}$ | h | 0.6 |
| Rate constant for reaction $E20$ | $k_{15}$ | 0.000013 | $yr^{-1}$ | g | 0.000013 |
| Rate constant for reaction $E21$ | $k_{16}$ | 0.0005 | $mmol^{-1}\ L\ yr^{-1}$ | g | 0.0027 |
| Rate constant for reaction $E22$ | $k_{17}$ | 0.025 | $mmol^{-1}\ L\ yr^{-1}$ | g | 0.052 |
| Rate constant for reaction $E23$ | $k_{18}$ | 0.000005 | $mmol^{-1}\ L\ yr^{-1}$ | - | Model constrained |
| Rate constant for reaction $E24$ | $k_{19}$ | 0.025 | $mmol^{-1}\ L\ yr^{-1}$ | - | Model constrained |
| Linear adsorption coefficient for $NH_4^+$ | $K_{NH4}$ | 1.3 | - | j | 1.3 |

[a]Arndt et al. (2013; based on their Fig.14 and the sedimentation rate and water depth of the Bornholm Basin); [b]Reed et al. (2011b); [c]Moodley et al. (2005); [d]Wang and Van Cappellen (1996); [e]Reed et al. (2011a); [f]Reed et al. (2016); [g]Egger et al. (2016); [h]Berg et al.(2003); [i]Egger et al. (2017); [j]Mackin and Aller (1984)

# Figures

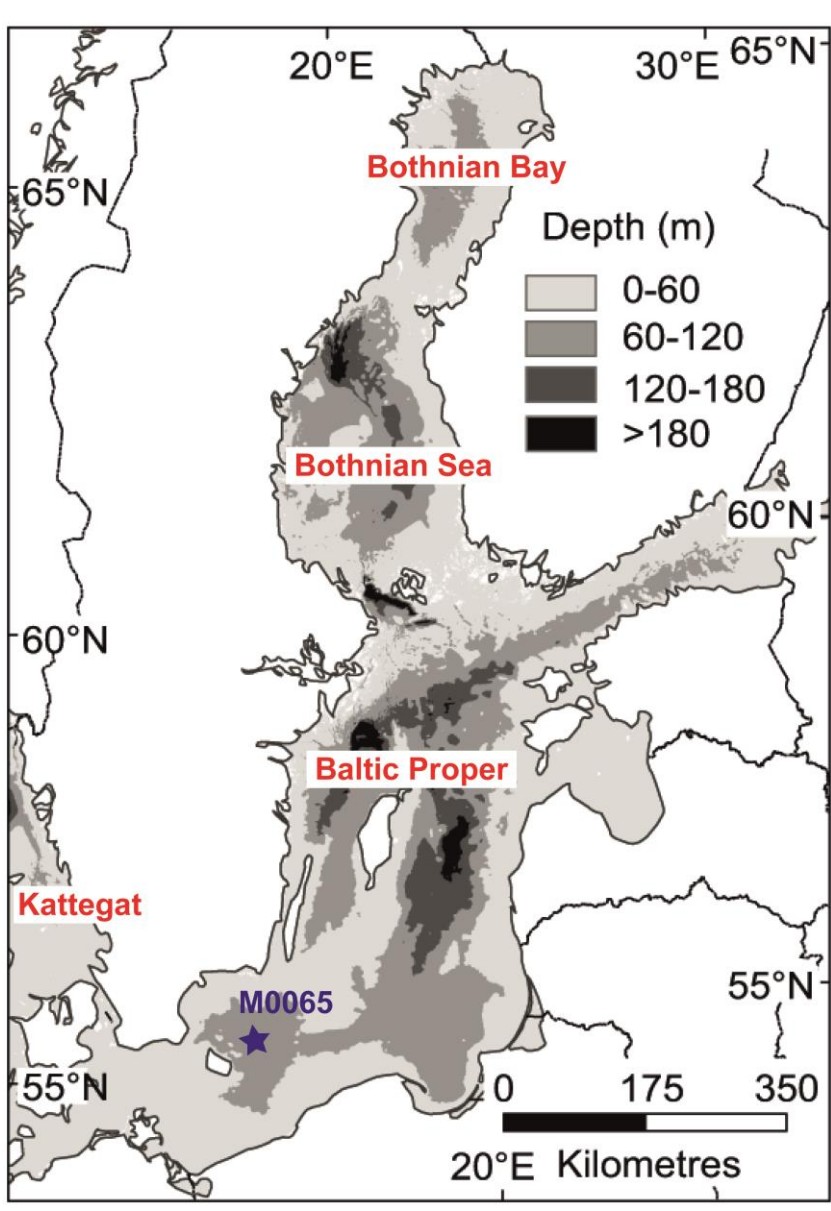

**Figure 1: Map of the major basins in the Baltic Sea with our study site (M0065) in the Bornholm Basin.**



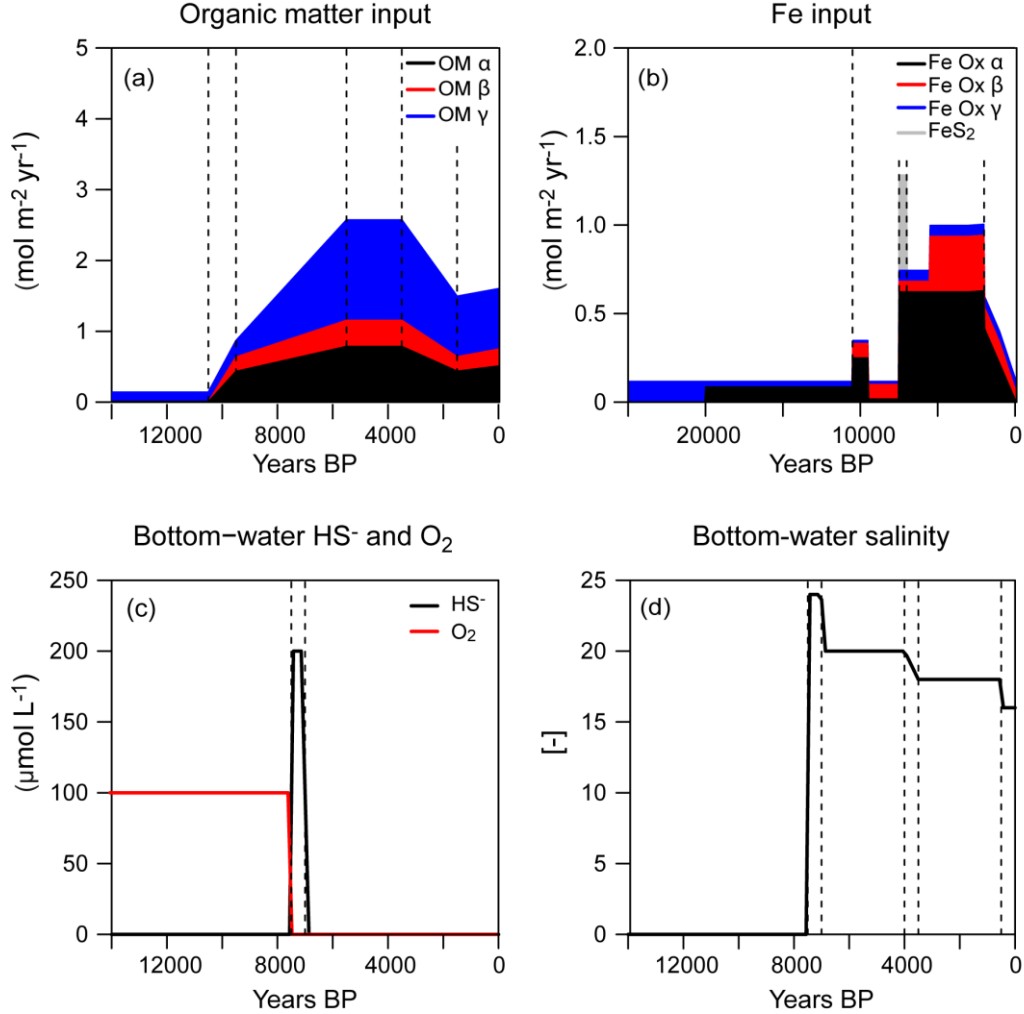

**Figure 2: Model scenario of organic matter (OM) input (a), Fe input (b), bottom water HS⁻ and O₂ concentrations (c), and bottom water salinity (d). Both Fe-oxides and organic matter were divided into highly reactive (α), less reactive (β) and inert (γ) phases. Note the different time scale for b. Inputs and bottom-water concentrations were kept constant before the plotted time interval.**



**Figure 3: Porewater trends with sediment depth for station M0065 (Holes A-C), including porewater data of the multicore from the same location (MC). Grey lines show profiles derived from the diagenetic model. The lake-marine (L-M) transition from Ancylus lake sediments to the brackish-marine Littorina Sea sediments is indicated in pink (brackish-marine sediments in green). The depth at which a large P enrichment is observed in the sediments (> 50 µmol/g) is shaded in brown (P peak).**







Figure 4: **Key elements at station M0065 with sediment depth (solid grey lines: model outputs). More details on the colored areas can be found in the legend of Fig. 3. Samples "Hole $A_{HR}$" were taken from Hole A cores more than one year after storage. *The $C_{org}$/P ratios were calculated using $C_{org}$ contents and estimated values of total P by interpolation between the nearest samples.**



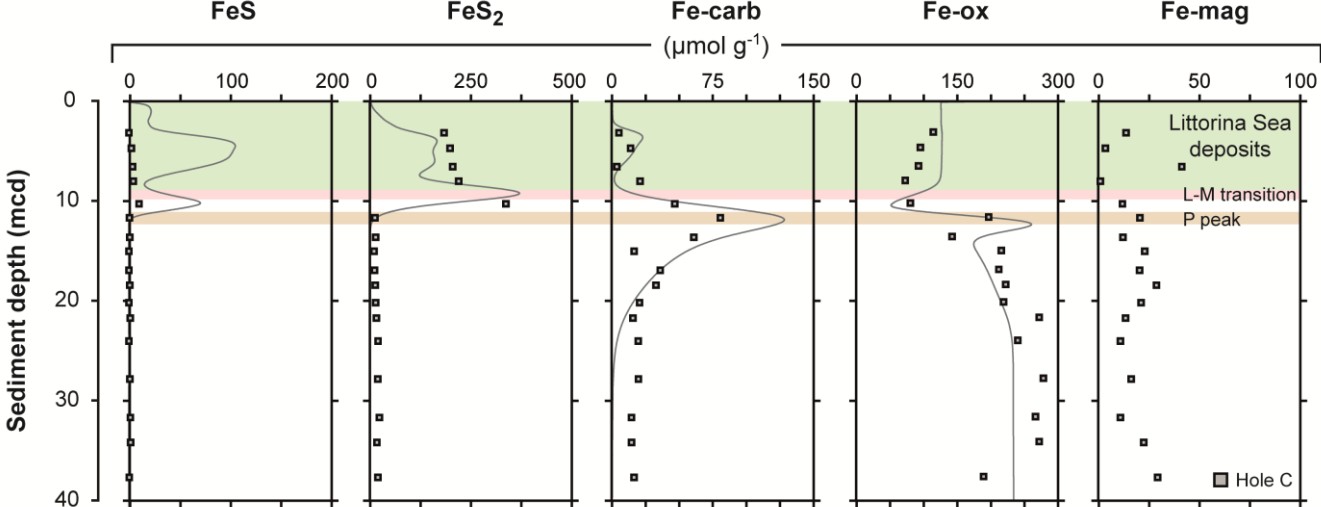

**Figure 5: Trends in S and Fe phases with sediment depth for Hole C. The S phases were extracted according to the procedure of Burton et al. (2008). Solid grey lines illustrate profiles derived from the diagenetic model. The Fe extraction of Poulton and Canfield (2005) was applied to determine Fe-carbonates (Fe-carb), sedimentary Fe-oxides (Fe-ox; sum of amorphous and more crystalline Fe-oxides) and magnetite (Fe-mag). Details on the colored areas can be found in the legend of Fig. 3.**





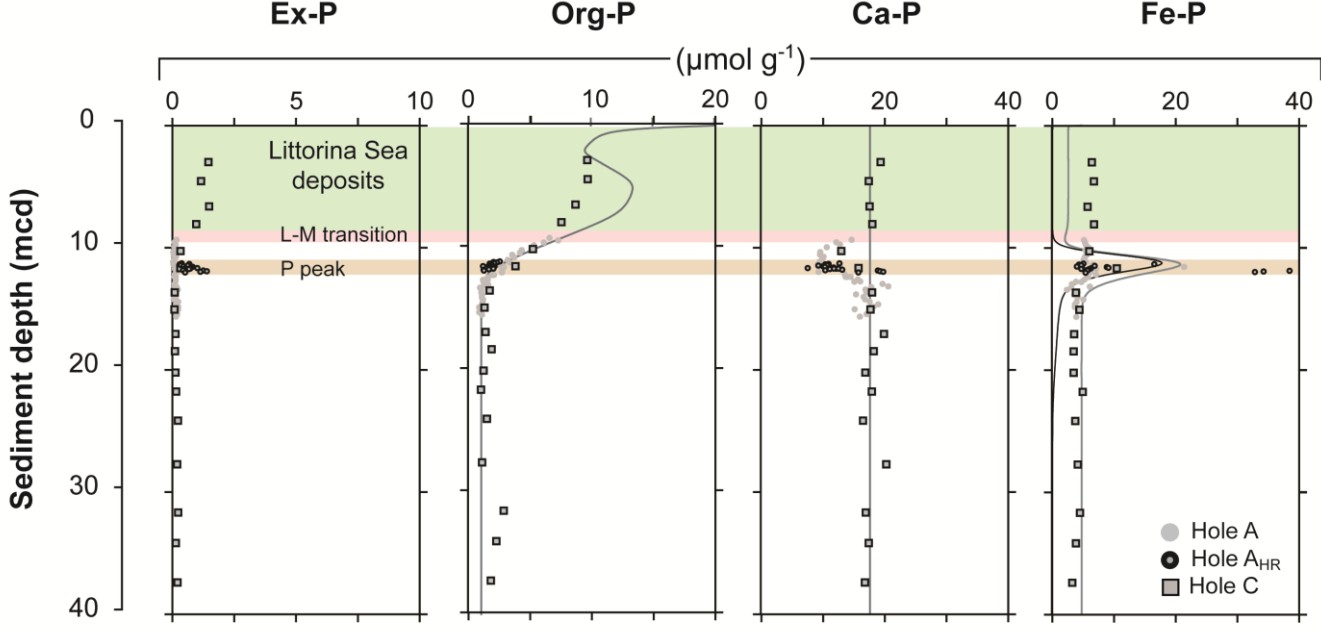

**Figure 6: The phosphorus fractionation in the sediments for Hole A and C, including exchangeable P (Ex-P), organic P (Org-P), authigenic and detrital Ca-P (Ca-P) and Fe-bound P (Fe-P). Grey lines show profiles from our model (vivianite trend in black). These phosphorus phases were determined according to the P extraction method of Ruttenberg (1992), including modifications by**

5   **Slomp et al. (1996). Samples "Hole $A_{HR}$" were taken from Hole A cores more than one year after storage. More details on the colored areas can be found in the legend of Fig. 3.**





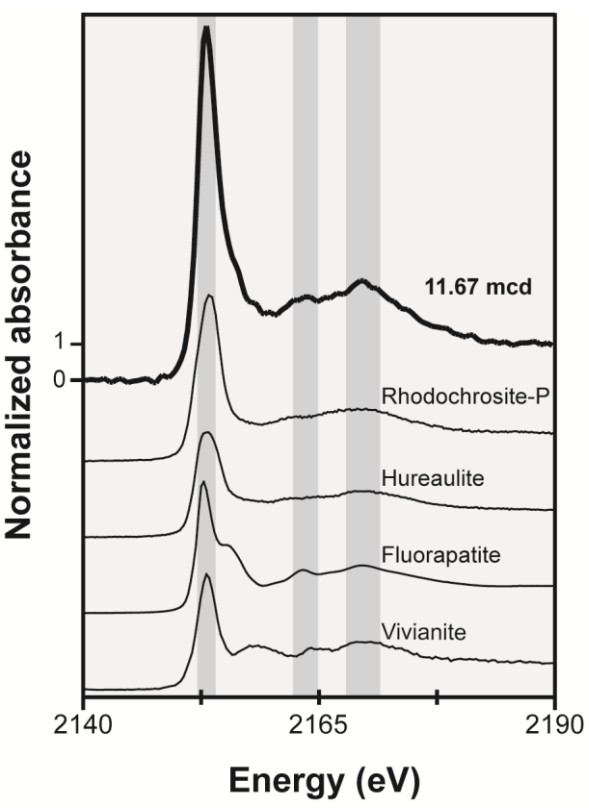

**Figure 7: Unfocused P-XANES spectra of bulk sediment from 11.67 mcd (Hole C) and various standards. All standards are described in Dijkstra et al. (2016). The dark grey shading highlights areas of specific interest (i.e. the position of white line and post-edge oscillations).**




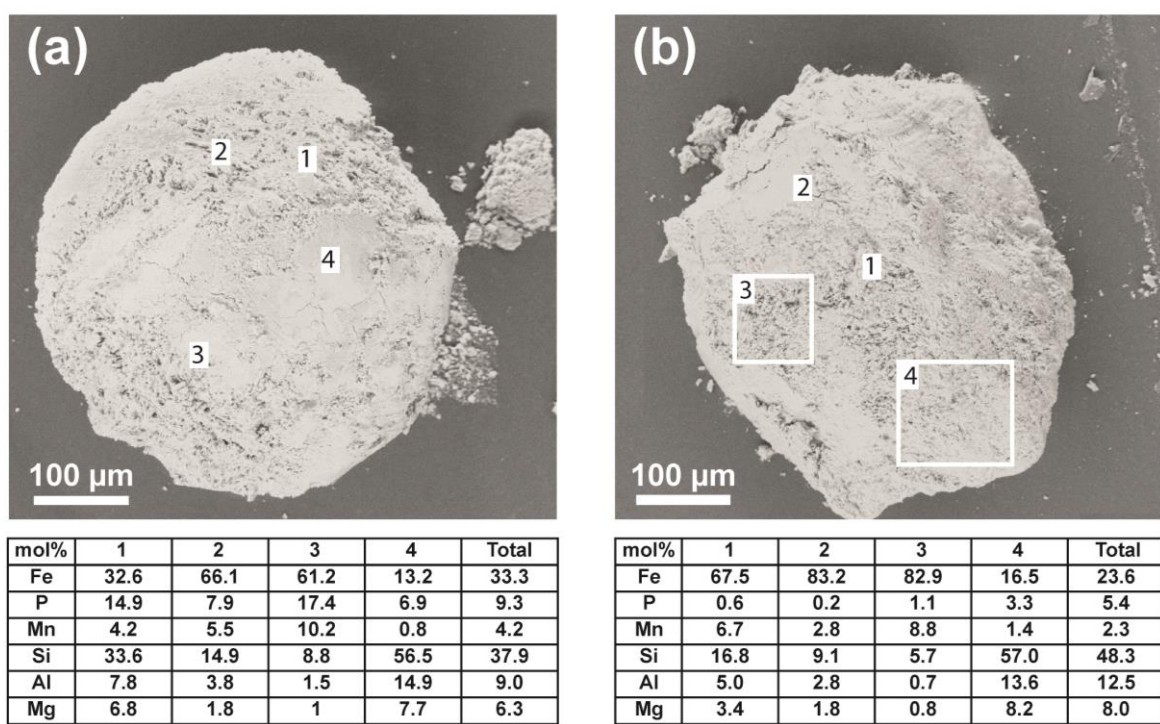

**Figure 8: SEM-EDS images of blue aggregates from 11.98 mcd (Hole $A_{HR}$) with elemental concentrations in mol%. The EDS results include analysis of spots and specific areas (1-5), and total area analysis.**





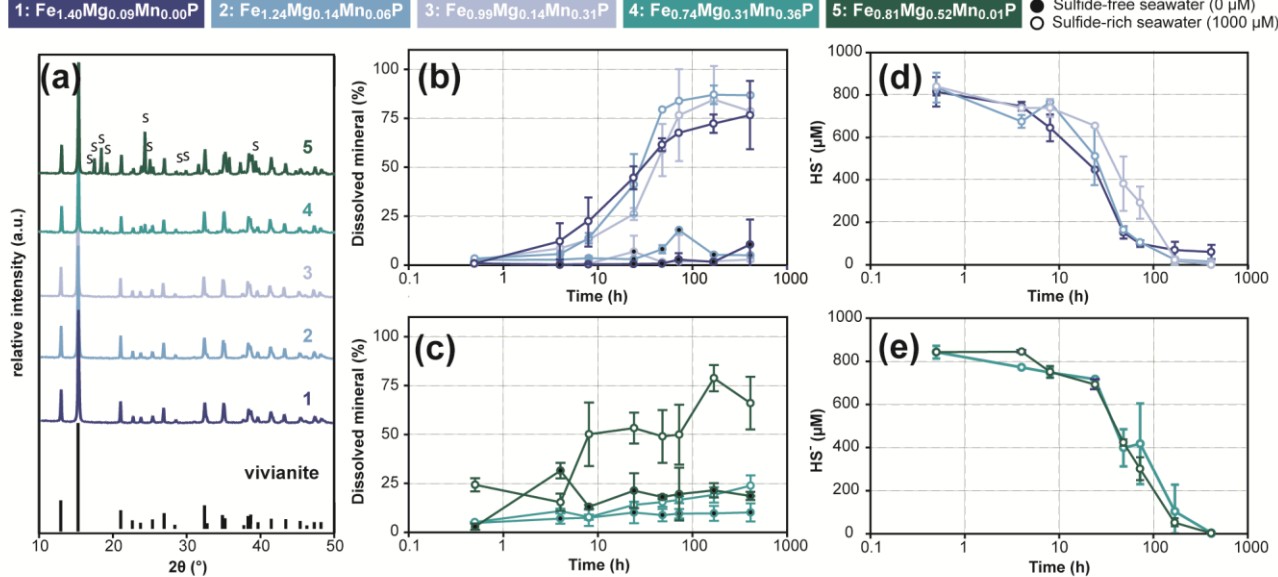

**Figure 9: Results from the vivianite sulfidization experiment including the XRD spectra of the precipitates and vivianite (A), the percentage of dissolved phosphorus (P) mineral (B, C) and the sulfide (HS⁻) concentrations in the sample solutions (D, E). Error bars reveal standard deviations (n=2) and sampling time is shown on a logarithmic scale. The XRD spectra of precipitate 4 and 5 also contained diffraction peaks that are indicative for struvite (S). The sulfide concentrations in the samples with initially 0 µM were all low (< 25 µM) and therefore not shown in the figure.**

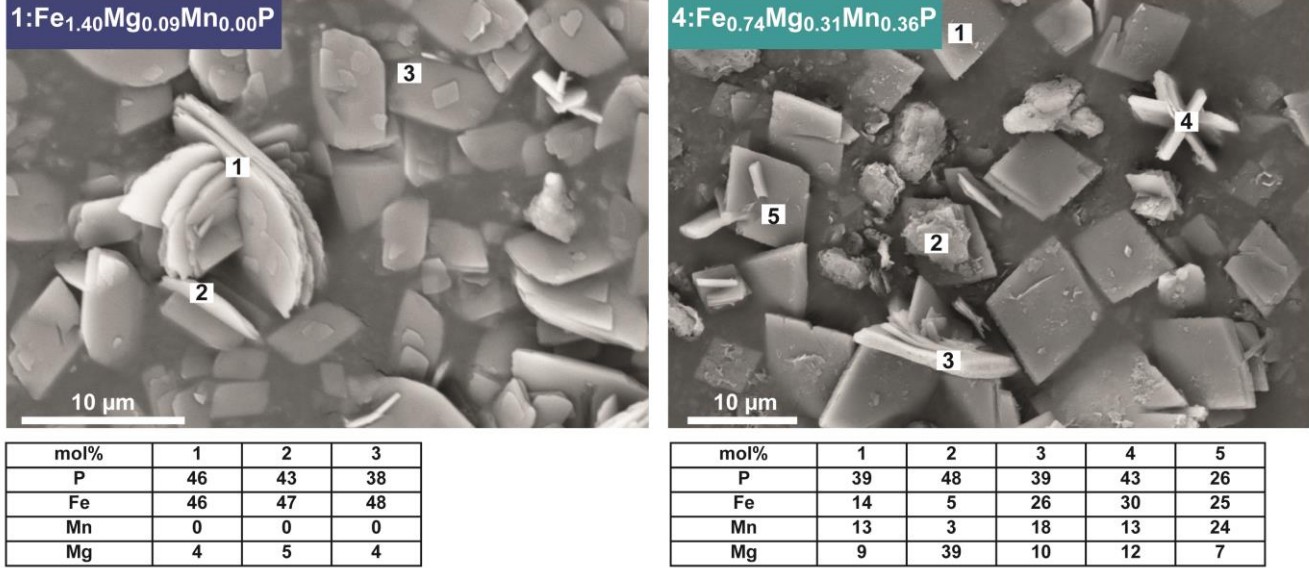

| mol% | 1 | 2 | 3 |
|------|----|----|----|
| P | 46 | 43 | 38 |
| Fe | 46 | 47 | 48 |
| Mn | 0 | 0 | 0 |
| Mg | 4 | 5 | 4 |

| mol% | 1 | 2 | 3 | 4 | 5 |
|------|----|----|----|----|----|
| P | 39 | 48 | 39 | 43 | 26 |
| Fe | 14 | 5 | 26 | 30 | 25 |
| Mn | 13 | 3 | 18 | 13 | 24 |
| Mg | 9 | 39 | 10 | 12 | 7 |

**Figure 10: Examination of precipitate 1 and 4 at the start of the sulfidization experiment by SEM-EDS, including imaging and elemental concentrations in mol%. Precipitate 1 generally consisted of ~ 5 µm crystals that contain phosphorus (P), iron (Fe) and magnesium (Mg). Precipitate 4 consists of a mixture of crystals rich in P, Fe, manganese (Mn) and Mg, and irregular shaped precipitates that contain both Mg and P. The precipitates also contained some sodium, chloride and sulfur (data is given in Supplementary Table S1).**





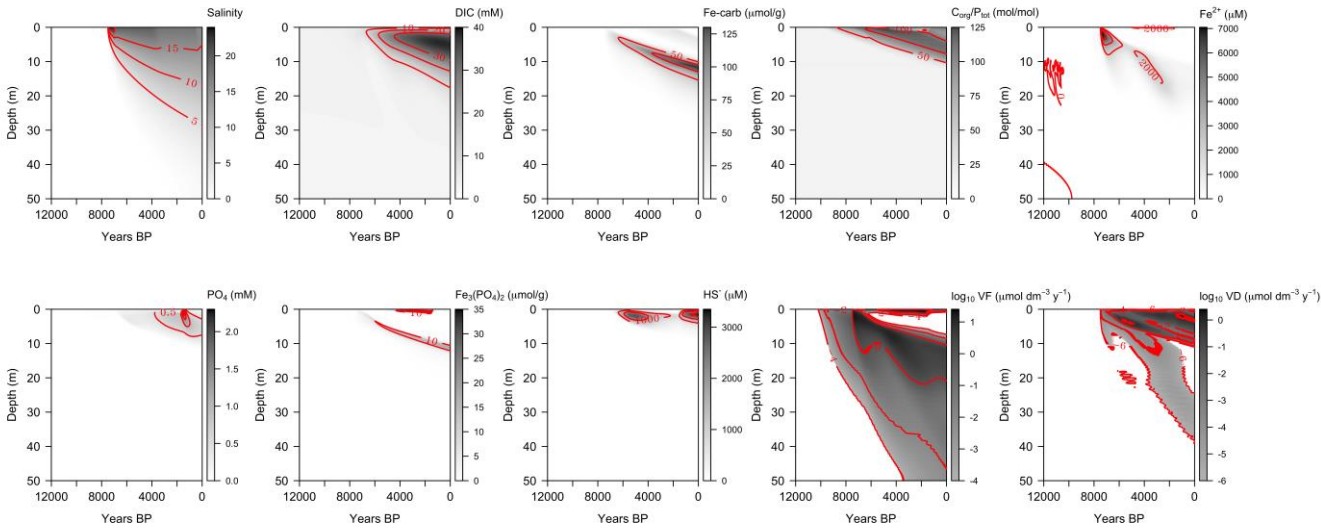

**Figure 11: Modeled evolution of salinity, dissolved organic carbon (DIC), Fe-carbonates (Fe-carb), $C_{org}/P_{tot}$, dissolved Fe ($Fe^{2+}$),**

**phosphate ($PO_4$), vivianite ($Fe_3(PO_4)_2$). dissolved sulfide ($HS^-$), vivianite formation (VF) and vivianite dissolution (VD) in the**

**Bornholm basin sediments. White color in the plots of VF and VD indicates values below the scale bar minimum.**





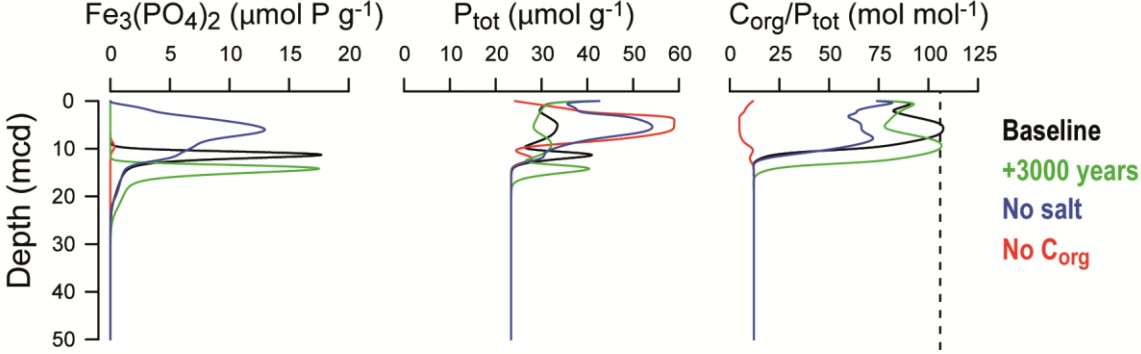

**Figure 12: Sensitivity of vivianite, total P concentration and the C$_{org}$:P-ratio in the sediments to changes in salinity and**

5    **productivity from 12.000 years B.P. to present, as well as 3000 years in the future with the current parameterization.**




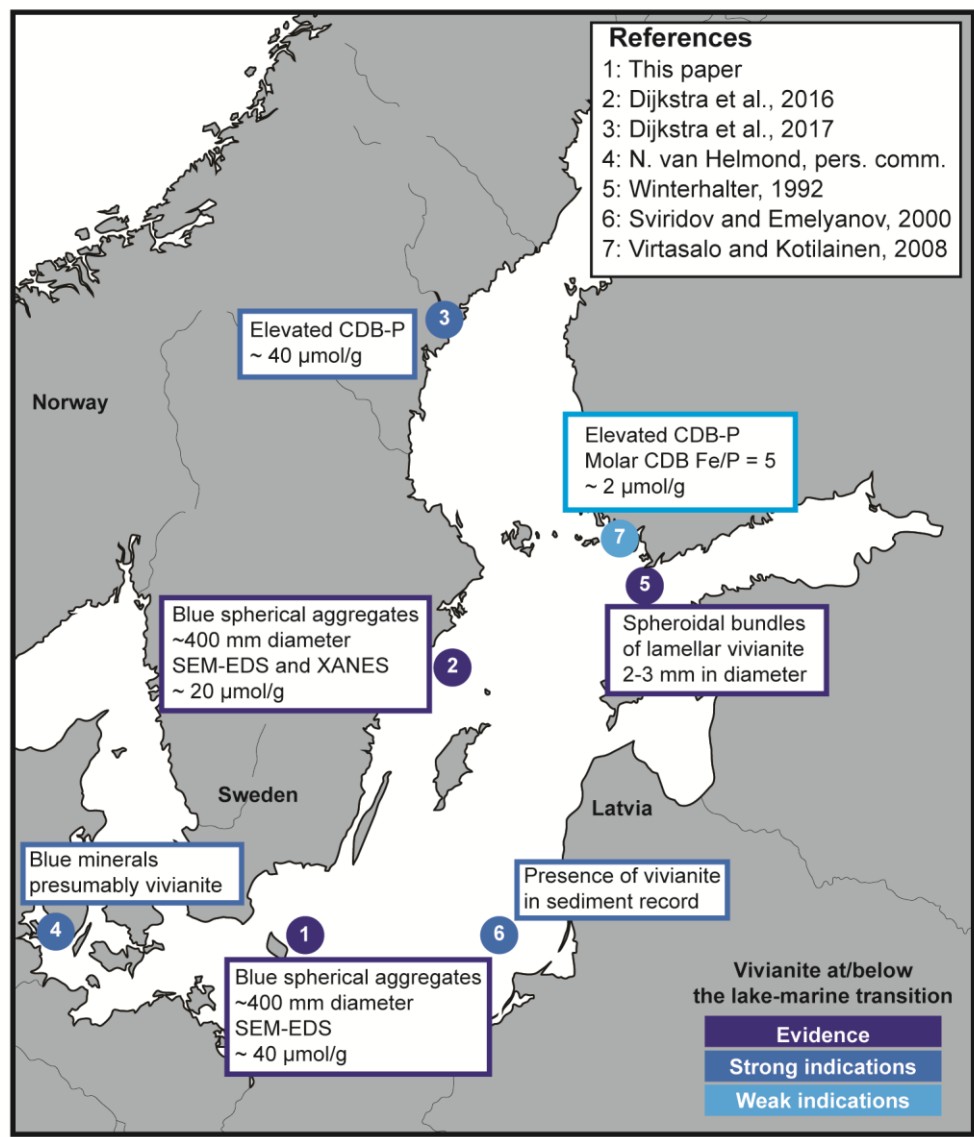

**Figure 13: Locations in the Baltic Sea where vivianite is or might be present at or below the lake-marine transition from Ancylus lake sediments to Littorina Sea sediments (see sect. 4.4 for further explanation). Phosphorus (P) forms that dissolve in a citrate-dithionite-bicarbonate extraction (CDB-P) are generally assumed to represent iron(Fe)-bound P.**




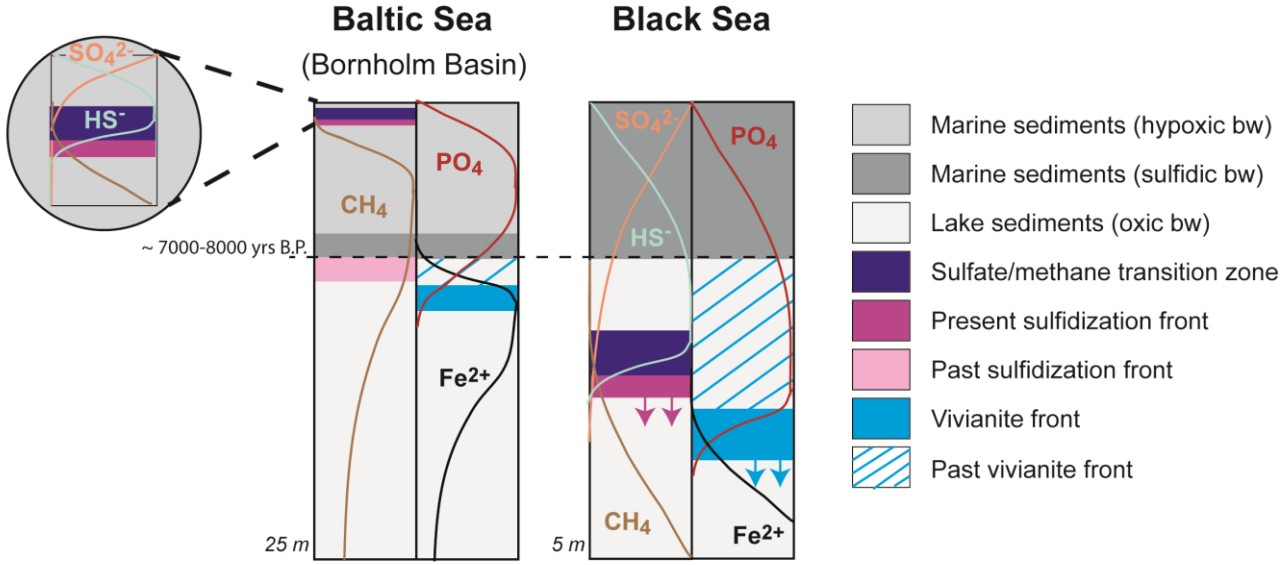

**Figure 14: Simplified schematic of methane (CH₄), sulfur (S), iron (Fe) and phosphorus (P) diagenesis in Baltic Sea (Bornholm Basin)**

**and Black Sea sediments (see sect. 4.4. for further explanation). Whereas the sedimentary vivianite peak in the Bornholm Basin is**

5   **currently stable with respect to its position relative to the lake-marine transition, the peak in post-depositional vivianite in the Black**

**Sea sediment is still migrating downwards from its initial location at the lake-marine transition (see sect. 4.4). The formation of**

**vivianite acts as a major sink for PO₄ in the lake sediments whereas the trend in porewater Fe²⁺ is affected mostly by upward**

**diffusion and, for the Black Sea, also by removal with sulfide. The vivianite diagenesis in deep basin sediments of the Black Sea is**

**discussed in detail by Egger et al. (2016). Note the difference in sediment depth between the Bornholm Basin and the Black Sea**

10   **sediment record.**