# Peer review of "Post-depositional formation of vivianite-type minerals alters sediment"

_Biogeosciences, 2017_

## Referee Comment (RC1) · Anonymous Referee #1 · 29 Oct 2017

This is a well-written paper presenting a large data set from cores taken in the Baltic Sea. The take home message that post-depositional vivianite formation can confound sediment P records is reasonable. The good news in terms of sedimentary P records is that vivianite forms and is mobilized under fairly unique conditions. Thus, especially in records where one sees fresh water to marine transitions, one should be mindful of potential alterations involving iron phosphates. I am not sure how common such transitions are in the geologic record. Nevertheless, it is good to be aware of this potential complication.

Perhaps I sense some frustration of the authors in this manuscript in that their studies yielded no direct evidence of the mineral vivianite in their system. Essentially the presence is inferred from extractions, modeling, blue particles and its presence in other

similar marine systems. The XANES measurements were not consistent with the presence of vivianite and the molar ratios of iron to phosphorus in SEM-EDS analyses do not seem to be that close to the values expected for vivianite. I have no doubt that there is some form of iron phosphate in these sediments but it may not be vivianite. There are many different iron phosphate minerals. My guess is that the iron phosphates are mixture of a number of poorly crystalline iron phosphate minerals. Given the absence of clear and direct evidence for vivianite, I believe it is a bit bold to state that its presence is "demonstrated" (line 7, page 2). Rather, it would be more accurate to say vivianite presence is inferred. Although using the term vivianite is a nice shorthand, it would be more representative of the findings to say something like "iron phosphates" in the title and throughout the text.

Discussion of the blue aggregates could be expanded. Mole percent analyses are presented in Figure 8 but they are not deeply discussed. How do these mole percent values compare with vivianite? How do they compare with other possible iron phosphate containing minerals?

Overall this is a very nicely presented study, other than the overly bold assertion for the specific presence of vivianite.

Minor issues:

Figure 11 is hard to understand. The graphs are tiny and it is unclear what the all the lines and shadings represent. Either the figure should be redesigned or a more extensive caption is needed to help the reader.

Line 9 page 13 change "is" to "are"

---

## Referee Comment (RC2) · Anonymous Referee #2 · 8 Nov 2017

This work investigates the potential role of vivianite (an iron II- phosphate mineral) precipitation in altering the trends of sedimentary P, a commonly used paleo-productivity proxy. The samples were collected in the Bornholm Basin (Baltic Sea) in 2013. A combination of modeling, biogeochemical and electron microscopy analyses and experiments shows that manganese and molybdenum rich vivianite precipitation under sulfidic conditions can strongly alter P sedimentary records after their deposition, especially when environmental perturbations such as primary productivity changes associated to marine-lake transitions occur. The authors nicely summarized an intensive laboratory work and they present a coherent manuscript. The results are relevant because they provide new insights to the use of sedimentary solid phase phosphorus analysis to reconstruct paleo environments. Vivianite in particular, seems a potentially

useful proxy for the occurrence of freshwater – marine transitions in systems like fjords in other parts of the world during, for instance, the last glacial maximum. However, the authors showed that vivianite peaks in the sediment could be strongly affected by sulfidic conditions and the presence of Fe2+, resulting in concentration peaks not directly related to water column productivity but to diagenetic reactions. I only have a few comments after reading the manuscript. My concerns are mostly related to format and some passages that I found a little bit "obscure". In methods I think the redaction of the paragraph referred to P XANES analysis could be improved (page 16, lines 10-14). I found hard to follow the procedure, probably because I am not familiar with this particular technique. In the text the authors mention a "white line" that I cannot see in Fig. 7. There is a problem citing first Fig. 11 (page 17, line 15) and then figure 10 (page 21, line 14). In addition, I think that Fig. 11 is complex compared to the rather brief references to it within the text. Moreover, where did ages the authors mention came up? (e.g. Page 18, lines 3-6). It is not clear if the authors used an age model and how they derived it. I guess they probably used ages estimated in the IODPreport by Andrén et al. (2015) or maybe, they used a constant sedimentation rate to estimate a composite core depth – age relationship. I think the authors should clarify this!

———————————————————

---

## Author Comment (AC1) · 2 Dec 2017

First of all, we would like to thank the reviewer for his/her comments and suggestions. Our response can be found below.

comment: This is a well-written paper presenting a large data set from cores taken in the Baltic Sea. The take home message that post-depositional vivianite formation can confound sediment P records is reasonable. The good news in terms of sedimentary P records is that vivianite forms and is mobilized under fairly unique conditions. Thus, especially in records where one sees fresh water to marine transitions, one should be mindful of potential alterations involving iron phosphates. I am not sure how common such transitions are in the geologic record. Nevertheless, it is good to be aware of this potential complication. Perhaps I sense some frustration of the authors in this manuscript in that their studies yielded no direct evidence of the mineral vivianite in their system. Essentially the presence is inferred from extractions, modeling, blue particles and its presence in other similar marine systems. The XANES measurements were not consistent with the presence of vivianite and the molar ratios of iron to phosphorus in SEM-EDS analyses do not seem to be that close to the values expected for vivianite. I have no doubt that there is some form of iron phosphate in these sediments but it may not be vivianite. There are many different iron phosphate minerals. My guess is that the iron phosphates are mixture of a number of poorly crystalline iron phosphate minerals. Given the absence of clear and direct evidence for vivianite, I believe it is a bit bold to state that its presence is "demonstrated" (line 7, page 2). Rather, it would be more accurate to say vivianite presence is inferred. Although using the term vivianite is a nice shorthand, it would be more representative of the findings to say something like "iron phosphates" in the title and throughout the text. Discussion of the blue aggregates could be expanded. Mole percent analyses are presented in Figure 8 but they are not deeply discussed. How do these mole percent values compare with vivianite? How do they compare with other possible iron phosphate containing minerals? Overall this is a very nicely presented study, other than the overly bold assertion for the specific presence of vivianite.

Reply: We agree that no direct evidence of vivianite has been presented in this study. However, we note that all the evidence points towards the presence of vivianite and not to other P minerals. The blue colour of the crystals is very typical for vivianite minerals. In addition, the Fe-P aggregates were almost identical in shape, color and size as vivianite minerals synthesized by Zelibor et al. (1988). It is also not unusual for the Fe:P molar ratio of vivianite, as determined with EDS, to deviate from 1.5 as discussed in our earlier work (Dijkstra et al., 2016). A previous study in the Bothnian Sea (Egger et al., 2015), to which we also refer in the main text, has shown that P-XANES is not always a conclusive method to provide evidence for (or against) the presence of

vivianite in bulk sediments. Even in sediments in which the presence of vivianite was proven by XRD, bulk P-XANES spectra did not always show features characteristic for vivianite possibly due to interference with the sediment matrix.

To address the reviewer's comment, we now refer to these minerals as vivianite-type minerals and iron(II)-phosphates in the text.

We also added the following sentences to page 20 line 12: "(as proven by XRD) in the Bothnian Sea. Due to this lack of direct evidence for the presence of vivianite, we refer to the blue Fe-P aggregates as vivianite-type minerals."

We now expanded the discussion part on the EDS results of Fig. 8 (page 19 line 20-25): "The P content of the aggregates was low compared to the Fe content of the aggregates, resulting in Fe:P-ratios that are higher than the expected stoichiometric Fe:P-ratio of vivianite (> 1.9 versus 1.5 mol mol-1). A similar discrepancy was observed in vivianite aggregates (as proven by XRD) in Dijkstra et al. (2016) and might reflect a surface coating of Fe, Mn and/or Mg of the aggregates, as has been observed in cold-seep sediments (Hsu et al., 2014)."

We removed "clearly" from page 20 line 2.

Comment: Figure 11 is hard to understand. The graphs are tiny and it is unclear what the all the lines and shadings represent. Either the figure should be redesigned or a more extensive caption is needed to help the reader.

Reply: We redesigned figure 11 and extended the caption as suggested (see supplement to this comment).

The added text reads: "The increase in salinity at 7500 BP marks the onset of the lake-marine transition. The subsequent deposition of sediment in a brackish-marine environment and the related changes in the porewater chemistry led to major changes in solid phase chemistry as a function of sediment depth and time. The distinct bands of Fe-carbonate and vivianite ($Fe_3(PO_4)_2$) minerals formed in the sediments are particularly striking."

In section 3.6, we also added a reference to the online supplementary information with an overview of the present-day process rates as a function of sediment depth: "Present-day rates of processes as calculated in the model are given in the online supplementary information"

Comment: Line 9 page 13 change "is" to "are"

Reply: Change made.

Please also note the supplement to this comment:
https://www.biogeosciences-discuss.net/bg-2017-295/bg-2017-295-AC1-supplement.pdf

**Supplement:**

**Supplement**

[Figure]

**Figure 11: Modeled evolution of salinity, dissolved inorganic carbon (DIC), Fe-carbonates (Fe-carb), $C_{org}/P_{tot}$, dissolved Fe ($Fe^{2+}$), phosphate ($PO_4$), vivianite ($Fe_3(PO_4)_2$). dissolved sulfide ($HS^-$) and rates of vivianite formation (VF) and vivianite dissolution (VD) in the Bornholm basin sediments from 12000 BP to present. A white color in the plots of VF and VD indicates values below the scale bar minimum. The increase in salinity at 7500 BP marks the onset of the lake-marine transition. The subsequent deposition of sediment in a brackish-marine environment and the related changes in the porewater chemistry led to major changes in solid phase chemistry as a function of sediment depth and time. The distinct bands of Fe-carbonate and vivianite ($Fe_3(PO_4)_2$) minerals formed in the sediments are particularly striking.**

---

## Author Comment (AC2) · 2 Dec 2017

First of all, we would like to thank the reviewer for his/her comments and suggestions. Our response can be found below.

Comment: This work investigates the potential role of vivianite (an iron II- phosphate mineral) precipitation in altering the trends of sedimentary P, a commonly used paleo-productivity proxy. The samples were collected in the Bornholm Basin (Baltic Sea) in 2013. A combination of modeling, biogeochemical and electron microscopy analyses and experiments shows that manganese and molybdenum rich vivianite precipitation under sulfidic conditions can strongly alter P sedimentary records after their deposition,

[Figure]

especially when environmental perturbations such as primary productivity changes associated to marine-lake transitions occur. The authors nicely summarized an intensive laboratory work and they present a coherent manuscript. The results are relevant because they provide new insights to the use of sedimentary solid phase phosphorus analysis to reconstruct paleo environments. Vivianite in particular, seems a potentially useful proxy for the occurrence of freshwater – marine transitions in systems like fjords in other parts of the world during, for instance, the last glacial maximum. However, the authors showed that vivianite peaks in the sediment could be strongly affected by sulfidic conditions and the presence of Fe2+, resulting in concentration peaks not directly related to water column productivity but to diagenetic reactions. I only have a few comments after reading the manuscript. My concerns are mostly related to format and some passages that I found a little bit "obscure".

In methods I think the redaction of the paragraph referred to P XANES analysis could be improved (page 16, lines 10-14). I found hard to follow the procedure, probably because I am not familiar with this particular technique. In the text the authors mention a "white line" that I cannot see in Fig. 7.

Reply: We have now added the main features (post-edge oscillations, white line, and shoulder feature) of the XANES spectra of the bulk sediment to the caption of Fig. 7 (see supplement). We now refer to the features in Fig. 7 in the main text (page 16, line 11): "See Fig. 7 for positions of XANES features."

Text added to caption: "The positions of main features of the bulk sediment spectrum (the white line, post-edge shoulder and the post-edge oscillations) are indicated in the figure. The white line (main absorption step) is observed in all P spectra, whereas the shoulder features are only visible in the bulk sediment spectrum and the fluorapatite spectrum."

Comment: There is a problem citing first Fig. 11 (page 17, line 15) and then figure 10 (page 21, line 14).

Reply: We now discuss the results of Fig. 10 in section 3.5 (page 17, line 9-14): "Precipitate 1 and 4 were also examined by SEM-EDS before the start of the experiment (Fig, 10). Precipitate 1 generally consisted of $\sim 5\ \mu$m crystals that contain phosphorus (P), iron (Fe) and magnesium (Mg). Precipitate 4 consisted of a mixture of crystals rich in P, Fe, manganese (Mn) and Mg, and irregular shaped precipitates that contain both Mg and P."

Fig. 11 is cited on page 17, line 19. To avoid duplication, we now removed the following sentences from the subtext of Fig. 10: "Precipitate 1 generally consisted of $\sim 5\ \mu$m crystals that contain phosphorus (P), iron (Fe) and magnesium (Mg). Precipitate 4 consists of a mixture of crystals rich in P, Fe, manganese (Mn) and Mg, and irregular shaped precipitates that contain both Mg and P."

Comment: In addition, I think that Fig. 11 is complex compared to the rather brief references to it within the text.

Reply: We have now redesigned Fig. 11 to make it easier to see the main trends (see supplement to this comment). The caption is now also more extensive. We note that most of section 3.6 discusses the trends in Fig. 11.

Comment: Moreover, where did ages the authors mention came up? (e.g. Page 18, lines 3-6). It is not clear if the authors used an age model and how they derived it. I guess they probably used ages estimated in the IODP report by Andrén et al. (2015) or maybe, they used a constant sedimentation rate to estimate a composite core depth – age relationship. I think the authors should clarify this!

Reply: Unfortunately, no age model for this site was available at the time of writing of this manuscript. This is why we relied on previous dating of the lake-marine transition and used a constant sedimentation rate in our model. This is also explained in the text (page 14, lines 4-9): "The sedimentation rate for the marine phase was estimated at 1.6 mm/y and kept constant for the entire simulation period, as we have no information on the sedimentation rate in the lake phase. This rate

corresponds to a lake-marine transition at ∼7.5 kyr as estimated by Zillen et al. (2008)."

Please also note the supplement to this comment:
https://www.biogeosciences-discuss.net/bg-2017-295/bg-2017-295-AC2-supplement.pdf

**Supplement:**

Supplement:

[Figure]

**Figure 7:** Unfocused P-XANES spectra of bulk sediment from 11.67 mcd (Hole C) and various standards. All standards are described in Dijkstra et al. (2016). The dark grey shading highlights areas of specific interest. The positions of main features of the bulk sediment spectrum (the white line, post-edge shoulder and the post-edge oscillations) are indicated in the figure. The white line (main absorption step) is observed in all P spectra, whereas the shoulder features are only visible in the bulk sediment spectrum and the fluorapatite spectrum.

[Figure]

**Figure 11:** Modeled evolution of salinity, dissolved inorganic carbon (DIC), Fe-carbonates (Fe-carb), $C_{org}/P_{tot}$, dissolved Fe ($Fe^{2+}$), phosphate ($PO_4$), vivianite ($Fe_3(PO_4)_2$). dissolved sulfide ($HS^-$) and rates of vivianite formation (VF) and vivianite dissolution (VD) in the Bornholm basin sediments from 12000 BP to present. A white color in the plots of VF and VD indicates values below the scale bar minimum. The increase in salinity at 7500 BP marks the onset of the lake-marine transition. The subsequent deposition of sediment in a brackish-marine environment and the related changes in the porewater chemistry led to major changes in solid phase chemistry as a function of sediment depth and time. The distinct bands of Fe-carbonate and vivianite ($Fe_3(PO_4)_2$) minerals formed in the sediments are particularly striking.